# Learning from Preferences and Mixed Demonstrations in General Settings

## Abstract

Reinforcement learning is a general method for learning in sequential settings, but it can often be difficult to specify a good reward function when the task is complex. In these cases, preference feedback or expert demonstrations can be used instead. However, existing approaches utilising both together are either ad-hoc or rely on domain-specific properties. Building upon previous work, we develop a novel theoretical framework for learning from human data. Based on this we introduce LEOPARD: Learning Estimated Objectives from Preferences And Ranked Demonstrations. LEOPARD can simultaneously learn from a broad range of data, including negative/failed demonstrations, to effectively learn reward functions in general domains. It does this by modelling the human feedback as reward-rational partial orderings over available trajectories. We find that when a limited amount of human feedback is available, LEOPARD outperforms the current standard practice of pre-training on demonstrations and finetuning on preferences, as well as other baselines. Furthermore, we show that LEOPARD learns faster when given many types of feedback, rather than just a single one.

## 1 Introduction

Reinforcement Learning (RL) is a branch of machine learning where an agent learns a behavioural policy by interacting with an environment and receiving rewards. These rewards are determined by a reward function that mathematically encodes the objective of the agent. For real-world practical applications of RL, such as robotics or Large Language Model (LLM) finetuning, the specification of the reward function poses a difficult challenge. Two popular RL subfields try to solve this problem by leveraging human data in order to learn what the reward function should be, typically by optimising a parameterised function such as a neural network.

Inverse RL (IRL) utilises human-provided demonstrations of the correct behaviour and tries to learn a reward function for which only the demonstrations, or similar behaviour, are near-optimal (Ng et al., 2000; Ziebart et al., 2008; Wulfmeier et al., 2015). RL from Human Feedback (RLHF) presents the human with pairs of agent–behaviour examples. For each pair, the human decides which piece of behaviour is better, and the reward function is trained to re-produce this preference (Christiano et al., 2017). Both methods iterate between reward model and agent training. For more details on IRL and RLHF, see sections 2.1 and 2.2, respectively. For many applications it might be possible and desirable to generate and learn from both of these feedback types, rather than committing to a single one. The current standard approach is to first train on demonstrations and then finetune the resulting model with preferences (Ibarz et al., 2018; Palan et al., 2019; Bıyık et al., 2022). Some methods have been proposed to more effectively leverage the information encoded in both the preferences and demonstrations, but this is still largely ad-hoc or specific to certain domains (Krasheninnikov et al., 2021; Mehta & Losey, 2023; Brown et al., 2019). We discuss these methods further in section 2.3.

In an attempt to solve this problem for general domains—and for many types of feedback including preferences and demonstrations—Jeon et al. (2020) propose Reward-Rational Choice (RRC). This frames the human feedback data as Boltzmann-Rational choices according to a probability distribution which has been induced by some unknown true reward function. Learning the reward function can then be cast as a supervised learning problem where we try to replicate these choices. Unfortunately, RRC is often difficult to implement in practice. For example, in the case of demonstration

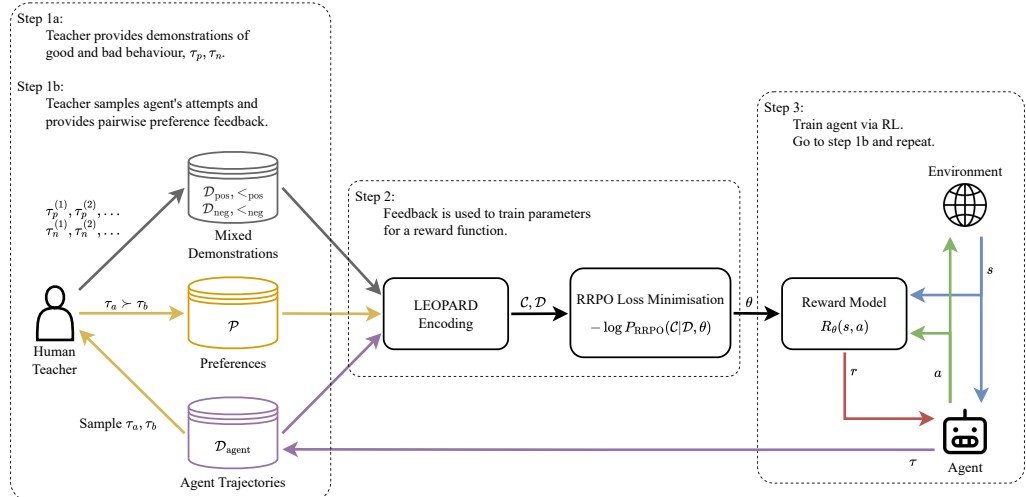

Figure 1: High-level overview of the LEOPARD algorithm. A teacher provides ranked examples of positive and negative demonstrations, as well as providing preference feedback over the agent's behaviour. This is used to train a reward model that the agent optimises via standard RL. The process is iterative. The LEOPARD encoding is given in Equations (7) and (8), and $P_{\text{RRPO}}$ is detailed in Equation (5).

feedback, they treat it as a choice over all possible behaviours. This space is incredibly difficult to optimise over if it is very large and our reward function is non-linear, as is often the case for practical problems. Additionally, it cannot encode multiple selections for the 'optimal choice', nor can it encode more complex relationships between behaviours such as rankings or dis-preference.

To address these limitations, we introduce a new theoretical framework which frames the human feedback as *reward-rational partial orderings* over trajectories (RRPO). These partial orderings are then encoded by sets of Boltzmann-Rational choices, analogous to the Plackett-Luce ranking model (Marden, 1996). From this we derive LEOPARD: Learning Estimated Objectives from Preferences And Ranked Demonstrations, which is outlined in Figure 1. In addition to preferences and ranked (positive) demonstrations, LEOPARD can also learn from ranked negative/failed demonstrations. Preferences are interpreted as they are in RRC, but positive demonstrations are interpreted as being preferred to the agent's current and future behaviour, or the opposite in the case of negative demonstrations. Demonstration rankings, if available, are also cleanly translated into partial orderings.

LEOPARD can utilise a wide range of feedback types simultaneously, making it effective at learning useful reward functions in general environments. We find that when preference and positive demonstration feedback is available, it outperforms the standard baseline of performing DeepIRL on the demonstration data, and then finetuning using preferences. It also beats Adversarial Imitation Learning with Preferences (AILP), another preference and positive demonstration learning algorithm, in three out the four environments tested on. Additionally, when only positive demonstration feedback is available, LEOPARD outperforms or matches DeepIRL and AILP due to its ability to exploit ranking data. Finally, we show that LEOPARD can learn more effectively when given a variety of feedback types, rather than focussing on large amounts of a single one.

To summarise, we make the following contributions:

1. We introduce RRPO, a practical and general framework for interpreting human feedback.

2. We introduce LEOPARD, an effective and scalable method for learning from preferences, and positive/negative ranked demonstrations.

3. We provide evidence that learning from many types of feedback can be superior to focussing on only one.

## 2 RELATED WORK AND BACKGROUND

### 2.1 DEMONSTRATION-BASED RL

A popular paradigm for learning from demonstrations is Inverse RL (IRL), where the demonstrations are used to learn a reward function (Ng et al., 2000). This overcomes many issues of behavioural cloning, which aims to directly mimic the given demonstrations (Bratko et al., 1995). Many current methods for IRL are based on the principle of *maximum (causal) entropy* (MaxEnt; MCE), established by Ziebart et al. (2008; 2010). This learns a reward function that captures the fact that the human demonstrations are optimal, but beyond this, it tries to have as much uncertainty about the reward dynamics as possible. Assuming a deterministic environment simplifies MCE into MaxEnt, and this assumption has been used to extend this class of methods into settings with high-dimensional observation spaces, e.g. DeepIRL (Wulfmuler et al., 2015). Advanced extensions of DeepIRL have been proposed, leveraging methods such as importance sampling (Finn et al., 2016), or GAN-style architectures (Fu et al., 2018). For a more comprehensive introduction to MCE and its derivatives, see Gleave & Toyer (2022). Our proposed algorithm does not reduce to a MaxEnt-derived method in the demonstration only case, but is still inspired by the principle and is of a similar form. Bayesian methods in IRL have also been explored (Ramachandran & Amir, 2007; Brown et al., 2020), highlighting how a probabilistic framing of the inverse learning problem can be useful.

### 2.2 PREFERENCE-BASED RL

RLHF (Christiano et al., 2017) use preferences—pairwise comparisons of agent behaviour—to learn a reward function for high-dimensional RL environments via the Bradley-Terry preference model (Bradley & Terry, 1952). A 3-step iterative procedure is used: sampling of new comparisons of recent agent behaviour, fitting the reward model to the comparison dataset, and training of the policy on the learnt reward function. The reward model is fitted by minimising the following loss function:

$$\mathcal{L}_{\text{RLHF}}(\theta) = - \sum_{(\tau_a, \tau_b) \in \mathcal{P}} \log P_{\text{RLHF}}(\tau_a \succ \tau_b | \theta), \tag{1}$$

where $\mathcal{P}$ is a dataset of pairs of trajectory-fragments[1] in which the first is preferred and

$$P_{\text{RLHF}}(\tau_a \succ \tau_b | \theta) = \frac{\exp(R_\theta(\tau_a))}{\exp(R_\theta(\tau_a)) + \exp(R_\theta(\tau_b))}, \tag{2}$$

where $R_\theta$ is a parameterised reward function. Wirth et al. (2017) provides a survey of other preference based RL methods prior to RLHF.

Recently, RLHF has been used for instruction and safety-finetuning large language models (LLMs) into chat systems (Ouyang et al., 2022; Bai et al., 2022; Bahrini et al., 2023). These are referred to as 'PPO-based' to disambiguate them from other methods which finetune LLMs from preferences without learning a reward function, such as DPO (Rafailov et al., 2024). Often the LLM is trained on demonstrations via behavioural cloning before PPO/DPO. Concerns for the safety, reliability, and misuse of LLMs has led to a plethora of research on how best to utilise human preferences/rankings to train these models (Cao et al., 2024; Chaudhari et al., 2024). Despite this, there is a broad lack of principled use of other feedback types for LLM safety and finetuning. Our method extends RLHF to be compatible with other sources of feedback, whilst still being practically applicable to problems like LLM finetuning.

### 2.3 COMBINING DEMONSTRATIONS AND PREFERENCE FEEDBACK

As mentioned in the case for LLMs, demonstration and preference feedback are typically combined by pre-training on the demonstration data using IRL/behavioural-cloning methods, and then fine-tuning the resulting reward model on preferences using RLHF (Ibarz et al., 2018; Palan et al., 2019; Bıyık et al., 2022). This works well in practice, but it is unclear how to add in further reward information, such as negative demonstrations or the relative rankings of demonstrations. Additionally,

---

[1]Contiguous subsequences of trajectories.

information that is present only in the demonstrations might be forgotten or never used, especially if strong regularisation is applied to the reward model, or the RL policy does not sufficiently explore when training on the demonstrations.

More sophisticated combinations of preferences and demonstrations have been considered. Krasheninnikov et al. (2021) sampled trajectories according to reward functions optimal for the preferences, and applied MCE-IRL. This approach is computationally expensive and limited to linear reward functions over tabular MDPs. Mehta & Losey (2023) combine preferences and demonstrations alongside corrections (Bajcsy et al., 2017), but leverage domain-specific properties of robotics and encode their demonstrations using trajectory-space perturbations. This method is not applicable outside of robotics, and loses information about how demonstrations are better than most of trajectory-space, not just better than nearby trajectories. Brown et al. (2019) and Brown & Niekum (2019) both subsample ranked demonstrations to produce preferences for training the reward model, giving good results but still losing information about how those demonstrations might be preferred to other trajectories. Taranovic et al. (2022) combines a novel preference loss with adversarial imitation learning. This is the closest to our work, and so we test against it as a baseline. We also note that none of these methods can be easily extended to other types of feedback.

Our method enables learning from preference and demonstration feedback in a principled manner, without leveraging domain-specific properties, and in a way that can be readily extended.

### 2.4 LEARNING FROM OTHER TYPES OF FEEDBACK

Other types of feedback have been explored in isolation, such as negative demonstrations (Xie et al., 2019),[2] improvements (Jain et al., 2015), off-signals (Hadfield-Menell et al., 2017a), natural language (Matuszek et al., 2012), proxy reward functions (Hadfield-Menell et al., 2017b), and even the initial state (Shah et al., 2019). Jeon et al. (2020) interpret many of these types of feedback as part of an overarching formalism, *reward-rational (implicit) choice* (RRC), providing a mathematical theory for reward learning that combines different types of feedback.

RRC interprets each piece of human feedback as a Boltzmann-Rational choice $C$ from some (possibly implicit) set of choices $\mathcal{D}$ with rationality coefficient $\beta$. A grounding function, $\psi$, maps choices to distributions over trajectories. The expected reward over these distributions gives the value for each choice under the Boltzmann-Rational model, according to some reward function $R_\theta$.

$$P_{\text{RRC}}(C|\mathcal{D}, \theta) = \frac{\exp\big(\beta \cdot \mathbb{E}_{\tau \sim \psi(C)}[R_\theta(\tau)]\big)}{\sum_{C' \in \mathcal{D}} \exp\big(\beta \cdot \mathbb{E}_{\tau \sim \psi(C')}[R_\theta(\tau)]\big)}. \tag{3}$$

For a deterministic $\psi$ this simplifies to:

$$P_{\text{RRC}}(C|\mathcal{D}, \theta) = \frac{\exp(\beta R_\theta(\psi(C)))}{\sum_{C' \in \mathcal{D}} \exp(\beta R_\theta(\psi(C')))}. \tag{4}$$

Many of the formalisms of feedback in RRC are not generally applicable, and practical applications rely on finite state-spaces or linear reward functions. For example, in the case of demonstrations it assumes access to the set of all possible trajectories, which is potentially uncountable and high-dimensional.

Our main theoretical contribution is adapting RRC to create RRPO, a more practical and expressive theoretical grounding of learning from general human feedback.

## 3 METHOD

We propose LEOPARD, a method for learning from preferences, positive demonstrations, negative demonstrations, and partial rankings over the given demonstrations. It is practical, flexible, and applicable to many environments. The aim is that a practitioner can give any and all feedback possible to the learning algorithm, and this feedback can be continuously learnt from and added to. First, we develop a general theoretical framework, reward-rational partial ordering (RRPO), extending that of deterministic reward-rational choice (RRC, Jeon et al. (2020)). Then, we apply this to the specific case of learning from preferences and mixed demonstrations.

---

[2]They refer to these as 'failed demonstrations'.

### 3.1 Reward Rational Partial Orderings

To ensure the general applicability of our theoretical formalisms, we assume that only the trajectories our reward optimisation procedure has access to are provided directly. These could be generated during the agent's training or provided by the human in the case of demonstrations. This is assumed as sensible/relevant trajectories could sit on an unknown manifold in (a high-dimensional) observation space, crippling random-sampling based approaches.[3] We'd expect that reward functions capturing complex desirable behaviour would not be linear, but that they could at least be approximated sufficiently by some differentiable parameterised function.

Our key insight is to interpret human feedback as a set of Boltzmann-Rational choices encoding strict partial orderings over the trajectory-fragments we have direct access to, where a fragment is a contiguous subsequence of a trajectory. For each item in the partial order, we 'choose' that element out of a set containing itself and all elements strictly less than it. This is analogous to the Plackett-Luce ranking model (Marden, 1996), and is equivalent when the ordering can be viewed as a total ordering embedded in some larger set. Similar to RRC, each partial ordering is assumed to be independent given the reward function. Since a partial order may encode a single element being greater than all others with no other relations, this generalises deterministic choices of RRC.

Formally, let $\mathcal{D} = \{\tau_i\}_i$ be the set of all possible fragments of trajectories we have access to, $\mathcal{C} = \{<_j\}_j$ the set of human feedback, and $R_\theta$ our non-linear reward function parameterised by $\theta$. Note that $<_i$ is used to denote some partial ordering $i$. We define the likelihood of $\theta$ under RRPO as follows:

$$P_{\mathrm{RRPO}}(\mathcal{C}|\mathcal{D},\theta) = \prod_{(\tau_i,<_j)\in\mathcal{D}\times\mathcal{C}} \frac{\exp(\beta_j R_\theta(\tau_i))}{\exp(\beta_j R_\theta(\tau_i)) + \sum_{\tau_k\in\mathcal{D}} \mathbf{1}_{\tau_k <_j \tau_i}\exp(\beta_j R_\theta(\tau_k))}, \quad (5)$$

where $\beta_j$ is the rationality coefficient for feedback $j$. $\beta$s should be equal if the type of feedback is the same, e.g. two pairwise preferences. Note that when the partial orderings are sparse, many terms of the product become unity. We perform gradient descent on the negative-log of eq. (5) to find the best $\theta$, giving the loss function below:

$$\mathcal{L}_{\mathrm{RRPO}}(\theta) = -\log P_{\mathrm{RRPO}}(\mathcal{C}|\mathcal{D},\theta). \quad (6)$$

A nice property of $\mathcal{L}_{\mathrm{RRPO}}$ is that when minimised it faithfully represents the partial orderings. More precisely, upper bounds on the loss give rise to lower bounds on all reward differences between fragments that are related by some partial ordering. This is stated formally and proved in theorem 1 of Appendix D. As a special case, if the loss is below $\log 2$ then all reward differences must have the correct sign, i.e. the reward function induces an ordering compatible with all the partial orderings.

### 3.2 LEOPARD

Whilst we can apply the framework above to many types of feedback, we now focus on the case of combining preferences with mixed demonstrations. By mixed demonstrations, we mean ones which may be positive, negative and, within these two groups, we may have access to the relative rankings of each demonstration.

A pairwise preference of $\tau_a \succ \tau_b$ is simply interpreted as a partial ordering with only $\tau_b < \tau_a$.[4] Positive demonstrations are interpreted as a single partial ordering that prefers all positive demonstrations to any agent trajectories and encodes the relative rankings of the positive demonstrations themselves. Negative demonstrations are interpreted likewise, but these partial orderings prefer agent trajectories over the negative demonstrations.

Formally, let $\mathcal{D}_{\mathrm{pos}}, <_{\mathrm{pos}}$, and $\mathcal{D}_{\mathrm{neg}}, <_{\mathrm{neg}}$ be the sets of trajectories and partial orderings encoding rankings from positive and negative demonstrations, respectively. Let $\mathcal{D}_{\mathrm{agent}}$ be the set of trajectories sampled from the agent's behaviour. Let $\mathcal{P} = \{(\tau_a, \tau_b)_i\}_i$ be the set of ordered pairs of trajectory-fragments in which the first is preferred, and $R_\theta$ our parameterised reward function. Then

---

[3]For example, consider the space of all images vs ones which are plausible 3D scenes.

[4]By interpreting each preference as its own partial ordering, we avoid potential issues of symmetry and non-transitivity.

we optimise the loss function, eq. (6), with the following:

$$<_{\text{Pos-Demo}} = <_{\text{pos}} \cup \{\tau_a < \tau_p | (\tau_a, \tau_p) \in \mathcal{D}_{\text{agent}} \times \mathcal{D}_{\text{pos}}\},$$
$$<_{\text{Neg-Demo}} = <_{\text{neg}} \cup \{\tau_n < \tau_a | (\tau_n, \tau_a) \in \mathcal{D}_{\text{neg}} \times \mathcal{D}_{\text{agent}}\},$$
$$\mathcal{C}_{\text{Pref}} = \{\{\tau_b < \tau_a\} | (\tau_a, \tau_b) \in \mathcal{P}\},$$
$$\mathcal{D}_{\text{pref}} = \bigcup_{(\tau_a, \tau_b) \in \mathcal{P}} \{\tau_a, \tau_b\},$$
$$\mathcal{C} = \{<_{\text{Pos-Demo}}, <_{\text{Neg-Demo}}\} \cup \mathcal{C}_{\text{Pref}}, \tag{7}$$
$$\mathcal{D} = \bigcup \{\mathcal{D}_{\text{pos}}, \mathcal{D}_{\text{neg}}, \mathcal{D}_{\text{agent}}, \mathcal{D}_{\text{pref}}\}. \tag{8}$$

Like in the case for RLHF, our dependencies on agent behaviour means we need to iterate between sampling new preferences, optimising for eq. (6), and training the agent's policy.[5] Our algorithm is illustrated in Figure 1 and the full training procedure is given in algorithm 1 in Appendix A, along with details on reward model training.

## 4 EXPERIMENTS

We test our method on several environments in order to evaluate its performance across a broad variety of domains. Additionally, we also vary the proportions and amounts of different types of feedback used for learning to demonstrate that combining demonstrations and preferences can give stronger performance than just relying on either one. In order to reduce the cost of testing our method and facilitate hyperparameter tuning with many repetitions, we synthetically generate preferences, demonstrations, and their rankings. We generate preferences by sampling using the sigmoid of the reward difference between the two fragments under comparison as the probability of preference. We generate demonstrations by training an agent on the ground truth reward function and then sampling its trajectories, with their ground truth reward determining their relative rankings. For further details, see Appendix A.2.

We experimentally evaluate LEOPARD on four environments from the Gymnasium (Towers et al., 2024) test suite: Half Cheetah (MuJoCo), Cliff Walking (Toy Text), Lunar Lander (Box2D), and Ant (MuJoCo). This covers a range of continuous and discrete observation and action spaces, reward sparsities, and overall complexities. We require a finite horizon to reduce complications from the preference and demonstration learning, so some environments required modification. These and other environment details are given in Appendix B.

We organise our experiments into two sections. In the first, we compare our method to baselines. For the case of preferences and positive demonstrations, we compare against Adversarial Imitation Learning with Preferences (AILP, Taranovic et al. (2022))[6] and a standard pipeline of training on demonstrations with DeepIRL and then preference finetuning with RLHF. As an ablation, on Half Cheetah we also test first training on preferences with RLHF, and then on demonstrations with DeepIRL. We find that, except on Ant, LEOPARD always outperforms all baselines. On Ant, it lags behind AILP but is still far better than the standard pipeline.

With positive demonstrations only, we show that LEOPARD either performs similarly or beats the baselines, depending on the environment, For preferences only, our method directly reduces to RLHF and so no comparison is needed. For LEOPARD and AILP, when training the reward model, we keep training until the loss has loosely converged (see Appendix A.1.3 for details). This is not possible with DeepIRL as the maximum-entropy 'loss' function is not bounded from below. Therefore, we use a fixed number of training epochs for the reward model with the associated baselines, and give results for a variety of values.

---

[5] If there were an existing set of preferences and agent trajectories, the method could be applied offline by simply optimising for eq. (6).

[6] For our implementation of AILP we only use the relevant loss functions and disregard the extraneous parts of the method. This includes initially optimising the policy to maximise visited state entropy, and sampling preferences according to maximum entropy. Additionally, we use PPO instead of SAC, and apply our early stopping method for reward model training. Overall this enables a fair comparison with LEOPARD, and we note that AILP's additional tweaks could be symmetrically applied to LEOPARD if so desired.

In the second set of experiments, we investigate how altering the types of feedback available affect reward learning performance. First, we see how well LEOPARD performs training only with preferences or only with positive demonstrations, choosing the amount available of each to be enough to enable learning but not enough to saturate performance. To enable a fair comparison, we normally choose equivalent amounts of feedback for the preference-only and positive-demonstration-only tests. For demonstrations, this is $n_{\text{demos}} \times$ trajectory-length, and for preferences this is $2 \times n_{\text{prefs}} \times$ preference-fragment-length.[7] Occasionally, one of the feedback types produced significantly worse performance, in which case we increased the proportion of feedback available. Details on trajectory and fragment lengths, feedback proportions, and other hyperparameters, are given in Appendix B.

Then, we test a 50/50 preferences / positive-demonstration mix, a 50/50 positive-demonstration / negative-demonstration mix, and a 50/25/25 preferences / positive-demonstrations / negative-demonstrations mix. These tests show that often mixtures of feedback types can outperform their single-typed counterparts, even when the total budget is fixed.

## 5 RESULTS

We present our results on how LEOPARD compares to common baselines, and how reward learning under our algorithm is affected by varying the types of feedback information.

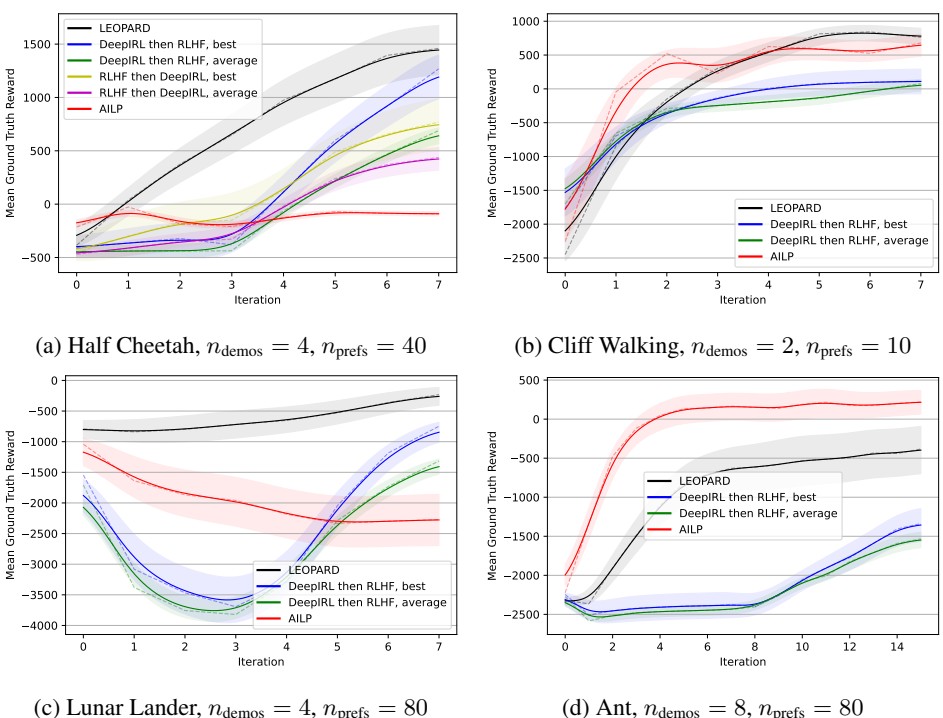

(a) Half Cheetah, $n_{\text{demos}} = 4$, $n_{\text{prefs}} = 40$     (b) Cliff Walking, $n_{\text{demos}} = 2$, $n_{\text{prefs}} = 10$

(c) Lunar Lander, $n_{\text{demos}} = 4$, $n_{\text{prefs}} = 80$     (d) Ant, $n_{\text{demos}} = 8$, $n_{\text{prefs}} = 80$

Figure 2: Comparison of LEOPARD with baselines of AILP, DeepIRL followed by RLHF, and RLHF followed by DeepIRL (Half Cheetah only), when positive demonstrations and preferences are available. The lines denote the mean of the ground truth reward function, with shaded standard errors, against algorithm iterations—alternations between optimising the reward model and the agent. Solid lines are smoothed means for clarity, dashed lines give raw values. A breakdown of the performance of the DeepIRL-based methods for different reward model training epochs per iteration is given in Figures 7 and 8.

Figure 2 compares LEOPARD to baselines when preferences and positive demonstrations are available, and Figure 3 analyses the case where only positive demonstrations are available. For a breakdown of individual final scores see Appendix C, Table 2.

---

[7] $\times 2$ as a preference involves comparing two fragments.

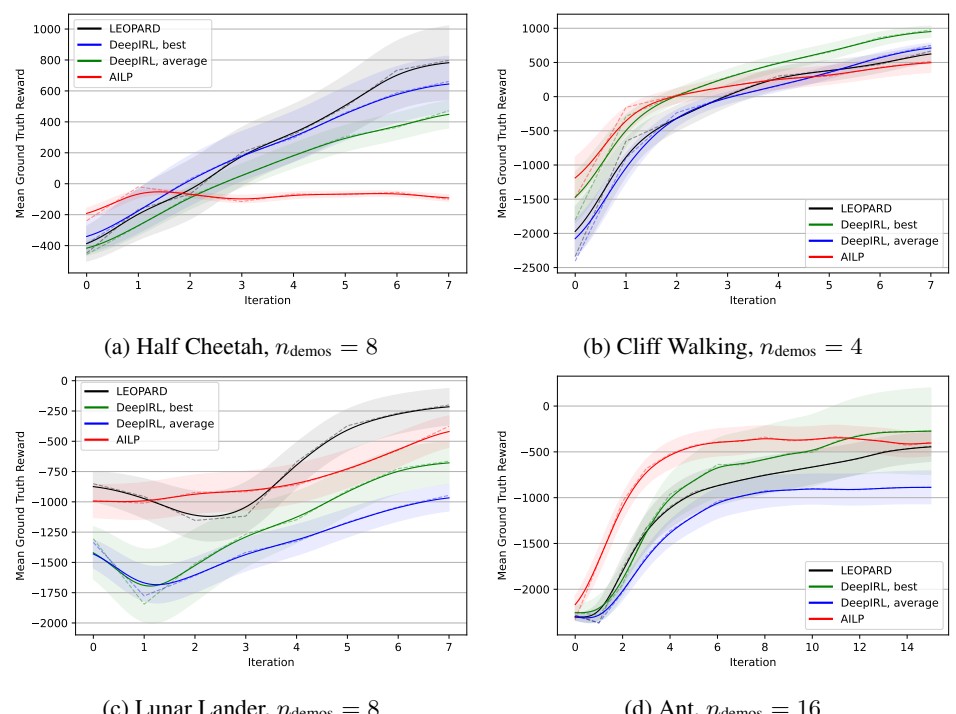

(a) Half Cheetah, $n_{\text{demos}} = 8$

(b) Cliff Walking, $n_{\text{demos}} = 4$

(c) Lunar Lander, $n_{\text{demos}} = 8$

(d) Ant, $n_{\text{demos}} = 16$

Figure 3: Comparison of LEOPARD with baselines of AILP and DeepIRL when only positive demonstrations are available. The lines denote the mean of the ground truth reward function, with shaded standard errors, against algorithm iterations—alternations between optimising the reward model and the agent. Solid lines are smoothed means for clarity, dashed lines give raw values. A breakdown of the performance of DeepIRL for different reward model training epochs per iteration is given in Figure 9.

We find that LEOPARD greatly outperforms the DeepIRL followed by RLHF baseline when both preferences and demonstrations are available, achieving much higher reward throughout training in all environments. Since LEOPARD can utilise all the data all the time, preferences can be used to aid early exploration, and demonstrations can continue to be trained against even in the latter stages. Additionally, as it trains the reward model to rough convergence each iteration it allows for adequate learning without over-fitting. It also beats AILP on three of the four environments, lagging slightly behind in Ant. Despite this, we still see LEOPARD as an improvement over AILP, since its performance with each iteration increases much more consistently. LEOPARD can exploit the relative rankings of the demonstrations to gain even more information on the underlying reward function compared to AILP, and the other baselines.

LEOPARD's use of ranking data and rough convergence training allows it to often outperform, and otherwise remain competitive with, the DeepIRL and AILP baselines when only demonstration data is available. We see a stronger relative performance on both Half Cheetah and Lunar Lander, whilst it is more clustered with the baselines on Cliff Walking and Ant. It is worth noting that LEOPARD does not require the 'reward model training epochs' hyperparameter, which might be difficult to tune for DeepIRL in environments that are expensive to sample from.

Note that for the analysis of the Cliff Walking environment, some outliers for the AILP[8] and 'DeepIRL then RLHF finetune' baseline have been removed. These were due to excessively large negative rewards from walking off the cliff many times before learning this was a bad idea, and occurred with an average frequency of 25% and 28% respectively. A more detailed breakdown along with the exact definition for outliers is given in Appendix C, Table 4.

---

[8]When training on preferences and demonstrations.

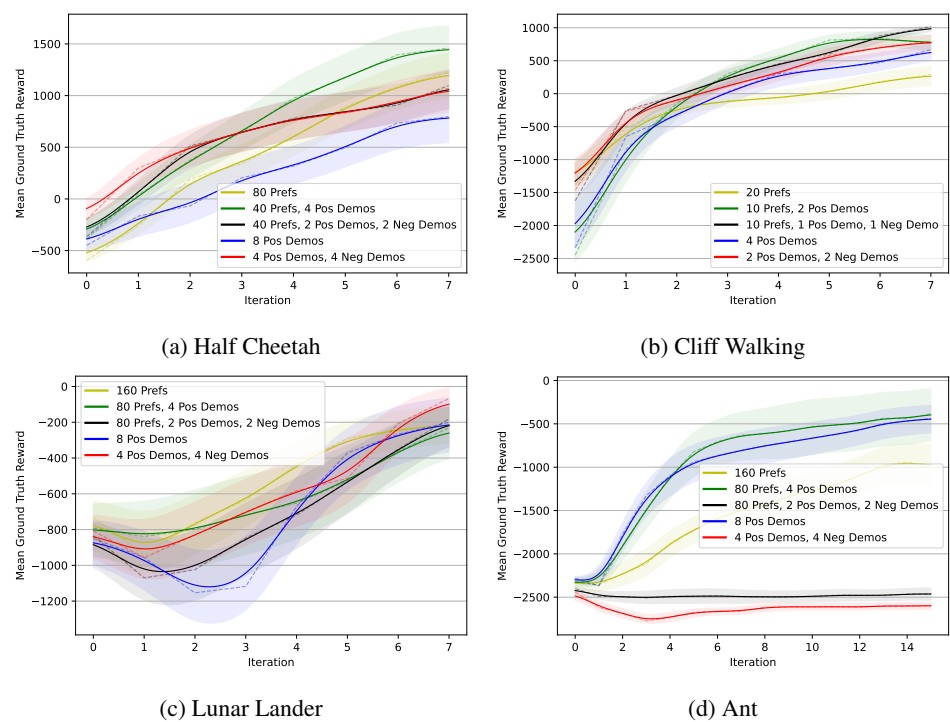

Figure 4: Comparison of LEOPARD's performance when varying types of feedback are available. The lines denote the mean of the ground truth reward function, with shaded standard errors, against algorithm iterations—alternations between optimising the reward model and the agent. Solid lines are smoothed means for clarity, dashed lines give raw values.

In Figure 4 we show the performance of LEOPARD when learning from a variety of different feedback proportions, with final scores detailed in Appendix C, Table 3. In all environments, some mix of preferences and demonstration data is top-scoring, and in two a pure feedback type is at the bottom. This is most clearly seen on Cliff Walking, where more diverse feedback types always beat their strict subsets. Interestingly, training only on preferences was better than using a full feedback mixture for the Half Cheetah environment, although a combination of preferences and positive demonstrations was much better than either. Whilst the mixed demonstrations strategy was the best for the Lunar Lander environment, the error bars there are large, and we caution against drawing clear conclusions. For Ant, both setups involving negative demonstrations did poorly, although the strongest performance was by preferences and positive demonstrations. We hypothesise that the poor performance of the negative demonstration containing runs might have been caused by limited representational capacity of the reward network to model three distributions of trajectory whilst still providing a useful feedback signal to the agent.

# 6 DISCUSSION

## 6.1 GENERALITY OF RRPO

Reward-rational preference orderings, the basis of LEOPARD, are a generalisation of the deterministic reward-rational choice framework (Jeon et al., 2020), but offers several distinct advantages. Recall that RRC frames the human feedback as a choice over some set, and then maps elements of that set into distributions over trajectories. Instead, RRPO maps the human feedback directly into a set of partial orderings. These two approaches have differing flexibility, and different feedback types might lend themselves more readily to one or the other. However, as RRPO is explicit in its construction that it operates only over directly-accessible trajectories, it becomes much more general in a practical sense. For example, in RRC, demonstration feedback requires optimising over the entire trajectory space, while RRPO does not.

Furthermore, RRPO does not assume any particular properties about the space of reward functions, nor the space of trajectories. In general, one can think of optimal trajectories as a small part of some feasible-trajectory manifold, which itself is a small part in a larger trajectory feature space. Methods which rely on domain-specific properties of these spaces, such as linearity or computable perturbations, inherently limit themselves from being more broadly applied. For example, Mehta & Losey (2023) leverages inverse kinematics models to interpret demonstration feedback (alongside preferences) in robotics domains. Whilst effective for this application, it renders the broader method impossible outside of robotics. RRPO and LEOPARD on the other hand, could be easily applied to environments very different to the ones we have tested on. For example, they could be used for Large Language Model (LLM) and foundation-model finetuning.

## 6.2 Limitations and Future Work

Whilst we have tested LEOPARD on a range of environments with differently structured observation and action spaces, a more comprehensive study would investigate an even wider range of tasks, such as more complex robotics, Atari games, and even LLM finetuning. Furthermore, with additional resources, it would be instructive to more closely interrogate how performance depends on the proportions of different feedback used for learning. For instance, future work could vary the feedback proportions with greater precision, and include additional repetitions.

Additionally, there are other methods that seek to learn from both preference and demonstration data, or even negative/failed demonstrations, as detailed in sections 2.3 and 2.4. Whilst these are less general in application than LEOPARD; a comparison of performance would still be interesting. We have chosen the baselines of AILP and 'DeepIRL followed by RLHF' to test against as they have similar simplicity and generality to our own method, as well as the latter being common practice.

We introduce RRPO as a theoretical backdrop for LEOPARD, however our investigation of its properties and encodings for many types of feedback is limited. Due to its similarity to RRC and the Placket-Luce choice model, we do not see this as a critical failing, as it will inherit many properties from those models, and deterministic RRC formulations can be trivially encoded under RRPO. Nevertheless, there are likely important theoretical properties and applications of RRPO that are of relevance to reward learning that ought to be investigated.

These limitations largely stem from constraints on time and computational resources. Thus, they are left to be resolved by future work.

## 6.3 Conclusion

We have shown that LEOPARD can perform effective reward inference, learning from many sources of reward information simultaneously. It is more effective than standard baselines for learning from preferences and demonstrations, and can additionally incorporate more information such as demonstration rankings and negative/failed demonstrations. We have also shown that using many sources of reward information can be more beneficial than relying on only large amounts of a single type. Whilst our empirical work is non-extensive, the generality and simplicity of the method makes it very powerful and potentially applicable to important current problems such as high dimensional robotics, and LLM / foundation-model finetuning. Furthermore, it opens the door to exploring the use of a much wider range of feedback in many RL settings.

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

## A  ALGORITHM DETAILS

The full algorithm for LEOPARD is given in algorithm 1. Initialisations follow standard neural network initialisation methods. RandomRollouts generates trajectories by sampling random actions and resetting the environment when necessary. TrainAgent performs standard PPO on the environment using the given reward function as the ground truth reward. Hyperparameters used for PPO are those given in RL Baselines3 Zoo (Raffin, 2020). Details on TrainRewardModel and GetPreferences are given in appendices A.1 and A.2.1 respectively. The generation of the demonstrations and their rankings is detailed in appendix A.2.2.

---

**Algorithm 1** LEOPARD

**Input**

| | |
|---|---|
| $n_{\text{iters}}$ | Number of iterations to perform |
| $n_{\text{rollout-steps}}$ | Number of environment rollout steps |
| $n_{\text{prefs}}$ | Number of preferences to sample |
| $\mathcal{D}_{\text{pos}}$ | Positive demonstrations |
| $<_{\text{pos}}$ | Positive demonstrations partial ordering |
| $\mathcal{D}_{\text{neg}}$ | Negative demonstrations |
| $<_{\text{neg}}$ | Negative demonstrations partial ordering |

**Output**

| | |
|---|---|
| $\pi$ | Trained agent policy |
| $R_\theta$ | Learnt reward function |

$n_{\text{rollout-steps-per-iter}} \leftarrow \lfloor n_{\text{rollout-steps}}/(n_{\text{iters}}+1) \rfloor$
$n_{\text{prefs-per-iter}} \leftarrow \lfloor n_{\text{prefs}}/n_{\text{iters}} \rfloor$
$\mathcal{D}_{\text{agent}} \leftarrow \emptyset$ $\qquad\qquad\qquad\qquad\qquad\qquad\qquad\qquad\qquad$ ▷ Agent trajectory pool
$\mathcal{P} \leftarrow \emptyset$ $\qquad\qquad\qquad\qquad\qquad\qquad\qquad\qquad\qquad\qquad$ ▷ Preferences dataset
$\pi \leftarrow \text{InitialiseAgent}()$
$R_\theta \leftarrow \text{InitialiseRewardFunction}()$
$\mathcal{D}_{\text{new-trajectories}} \leftarrow \text{RandomRollouts}(n_{\text{rollout-steps-per-iter}})$

**for** 1 to $n_{\text{iters}}$ **do**
$\quad \mathcal{P} \leftarrow \mathcal{P} \cup \text{GetPreferences}(n_{\text{prefs-per-iter}}, \mathcal{D}_{\text{new-trajectories}}, \mathcal{D}_{\text{agent}})$
$\quad \mathcal{D}_{\text{agent}} \leftarrow \mathcal{D}_{\text{agent}} \cup \mathcal{D}_{\text{new-trajectories}}$
$\quad R_\theta \leftarrow \text{TrainRewardModel}(R_\theta, \mathcal{D}_{\text{pos}}, <_{\text{pos}}, \mathcal{D}_{\text{neg}}, <_{\text{neg}}, \mathcal{D}_{\text{agent}}, \mathcal{P})$
$\quad \pi, \mathcal{D}_{\text{new-trajectories}} \leftarrow \text{TrainAgent}(\pi, R_\theta, n_{\text{rollout-steps-per-iter}})$
**end for**

---

### A.1  REWARD MODEL TRAINING

The reward model is trained by optimising the loss function eq. (6) with the AdamW optimiser. Batches of $\mathcal{D}_{\text{pos}}, \mathcal{D}_{\text{neg}}, \mathcal{D}_{\text{agent}}$, and $\mathcal{P}$ are sampled as detailed in appendix A.1.1, and then encoded via eqs. (7) and (8). Additionally, the batch loss is normalised according to the batch size, detailed in appendix A.1.2. Instead of training for a fixed number of steps / epochs, training steps are taken until some stopping condition is achieved, as detailed in appendix A.1.3. Together these procedures could result in varying coverages for each data source, from potentially many 'epochs',[9] to only sampling a small fraction of it.

### A.1.1  BATCH SAMPLING

$\mathcal{D}_{\text{pos}}, \mathcal{D}_{\text{neg}}, \mathcal{D}_{\text{agent}}$, and $\mathcal{P}$ are independent, heterogeneous, and in general of different sizes. This makes batch sampling non-trivial to perform. First batch sizes for each of the data sources is determined, and then each one is sampled independently. As is typical, they are sampled without

---

[9]Since our data sources are of varying sizes and not partitioned into equal numbers of batches, the notion of a training epoch - one complete pass over all training data - is not well-defined. We do however have notions of data source specific epochs.

replacement until empty, and then reset, potentially multiple times if that's required to fill the batch. Batches for $<_{\text{pos}}$ and $<_{\text{neg}}$ are simply derived from the respective batches of $\mathcal{D}_{\text{pos}}$ and $\mathcal{D}_{\text{neg}}$.

There is a maximum batch size for the trajectory-type data sources ($\mathcal{D}_i$), and a maximum batch size for $\mathcal{P}$. These could be different as trajectory-fragments are typically smaller than trajectories, and we may want to ensure a portion of (V)RAM is available for each. Batch sizes are also generated to be somewhat proportional to the size of their respective datasets. This is important as we don't want to diminish the importance of a data source that has lots of data generated for it, nor over-represent data sources with only a few data points. Once the proportionality constants are known, the sizes are scaled so that at least one of the batch sizes is at its maximum, and none of them exceed their maximums. Some data sources, namely $\mathcal{D}_{\text{agent}}$, are treated as 'in-excess', and not taken into account when trying to make batch sizes proportional to dataset sizes. These are simply given their maximum size.

### A.1.2   Loss Normalisation Across Batch

As we want our gradient steps to be roughly unity in magnitude and independent of the batch size, we need to normalise it. Typically, this is very easy in supervised learning—one can simply take an average across the batch—but this is not the case for eq. (6). Expansion of the gradient of the loss with respect to $\theta$, and noting our reward function operates at the level of transitions within trajectories, reveals the correct normalising factor to divide by:

$$\sum_{(\tau_i, <_j) \in \mathcal{D} \times \mathcal{C}} \text{Length}(\tau_i) \cdot \mathbf{1}_{\exists \tau_k \in \mathcal{D}. \tau_k \neq \tau_i \wedge \tau_k <_j \tau_i}.$$

This assumes a fixed length of fragments for each partial ordering.

### A.1.3   Stopping Conditions

Generally, the reward function loss from poorly-fitted demonstration rankings are much higher than poorly fitted preferences. This is because trajectories are typically longer than trajectory-fragments and demonstrations generate more '$<$' comparisons than a preference. However, the distribution of demonstrations are typically quite far from that of the agent trajectories, which the preferences have been generated over. This makes it much easier for the reward function to separate the demonstrations from agent behaviour and thus achieve a low loss on the demonstration ordering, than it does for it to get low loss on all the preference orderings.

The consequence of the above two facts is that if we were training on just the demonstrations, we'd want to do at most a few epochs (to learn fast and avoid overfitting), but if we were training on just the preferences we might want to do more (as learning is slower and overfitting less of a potential issue). Thus, as the amount of data in each dataset varies in each iteration, it does not make sense to have a pre-specified number of training steps, and instead a stopping condition should be used.

Our stopping condition simply checks if the training loss has loosely converged. At each step we check if the training loss is within $\pm 0.001$ of the last step's training loss. If this occurs 3 times in a row, we stop training the reward model for that iteration, and return to agent training. Empirically this strikes the balance between learning and avoiding overfitting.

### A.2   Synthetic Feedback

### A.2.1   Preferences

In algorithm 1, the GetPreferences function randomly samples trajectory fragments for comparison, with a bias to sampling from new trajectories. We are using a synthetic oracle which uses the ground truth reward function to noisily generate preferences, simulating the imperfect human rationality. More specifically, for each sampled pair of fragments, the sigmoid of their reward difference is used as the parameter for a Bernoulli random variable which is then sampled to generate the preference.

### A.2.2   Demonstrations

To create demonstrations for our tasks, we simply train an agent on the ground truth reward function (or its negation in the case of negative demonstrations). Several agents are trained, and the best

few, $n_{\text{selected}}$, are picked. From these agents, we create a list of their trajectories, ordering from their latest attempts to their first, and interleaving each agent together with the best agent first. For training an agent from feedback, if $n$ demonstrations are being used, the first $n$ demonstrations from this list are provided. Rankings are generated automatically based on the ground truth reward of each demonstration, making $<_{\text{pos}}$ and $<_{\text{neg}}$ total orders.[10] The ground truth reward per agent step and number selected, $n_{\text{selected}}$, of all demonstrations trained are given in Figures 5 and 6 for positive and negative demonstrations respectively.

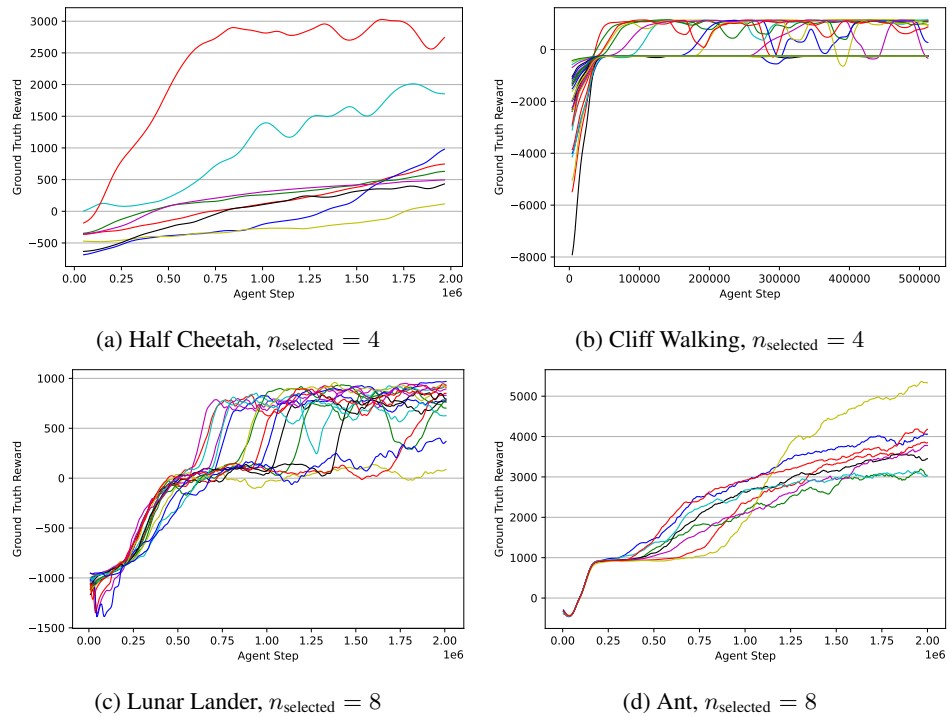

(a) Half Cheetah, $n_{\text{selected}} = 4$

(b) Cliff Walking, $n_{\text{selected}} = 4$

(c) Lunar Lander, $n_{\text{selected}} = 8$

(d) Ant, $n_{\text{selected}} = 8$

Figure 5: Ground truth reward vs agent steps for the positive demonstrations that were trained in every environment. We also state how many were selected as good examples to be used for demonstration learning.

---

[10]They are not required to be total orders to apply the general method.

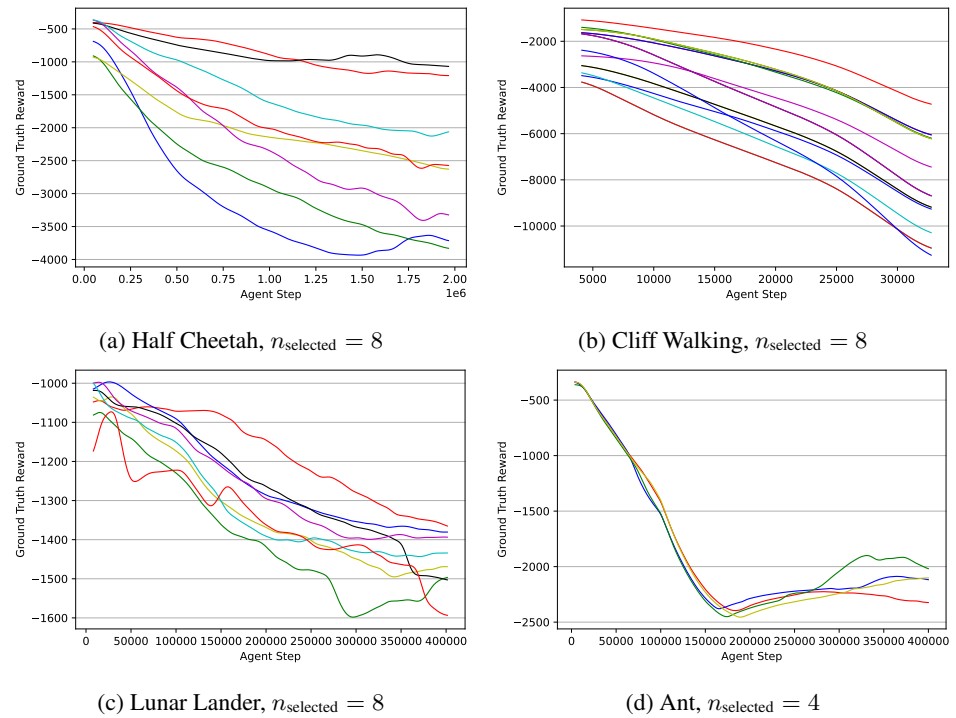

Figure 6: Ground truth reward vs agent steps for the negative demonstrations that were trained in every environment. We also state how many were selected as bad examples to be used for demonstration learning.

# B ENVIRONMENT DETAILS

Here we give details on versions / modifications made for each environment, as well as environment-specific hyperparameters summarised in table 1.

| Environment | Traj Len | Pref Len | $n_{\text{iters}}$ | $n_{\text{rollout-steps}}$ | Rng Seeds | Pref-time : Demo-time |
|---|---|---|---|---|---|---|
| Half Cheetah | 1k | 50 | 8 | 8M | 32 | 1:1 |
| Cliff Walking | 250 | 25 | 8 | 256k | 16 | 1:1 |
| Lunar Lander | 250 | 50 | 8 | 8M | 24 | 8:1 |
| Ant | 1k | 50 | 16 | 16M | 12 | 1:1 |

Table 1: Environment specific hyperparameters. 'Traj Len' refers to the fixed trajectory length for that environment, 'Pref Len' is the length of preference fragments - the contiguous trajectory subsequences that are used to generate preferences. Both are measured in environment timesteps.

## B.1 HALF CHEETAH

The v4 version is used out-of-the-box, trajectories are 1k timesteps and preference fragments are 50 timesteps. 8 iterations are used with a total of 8M environment rollout steps. Results are averaged over 32 different seeds.

## B.2 CLIFF WALKING

The v0 version is modified to have a fixed horizon of 250 timesteps and a custom reward function. Preference fragments are 10 timesteps, and 8 iterations are used with a total of 256k environment rollout steps. Results are averaged over 16 different seeds.

The standard version has a reward of -1 every timestep with the episode terminating when the end is reached. Walking off the cliff gives -100 reward and returns the agent to the start. Our fixed horizon version of this is the same except reaching the end state does not terminate the environment, and instead grants 5 reward per timestep spent there. This was based on what lead to good learning with PPO and access to the reward function directly.

As the reward function is sparse, for sampling preferences only, a shaped version of it is used to simulate human intuition on what behaviours are closer to optimal. The penalty for walking off cliffs remains the same, but otherwise the agent receives a weighted reward of -1 and 5 depending on how close in $L_1$ norm it is to the start/end state respectively.

### B.3  LUNAR LANDER

The v2 version is modified to have a fixed horizon of 250 timesteps and a custom reward function. Preference fragments are 50 timesteps, and 8 iterations are used with a total of 16M environment rollout steps. Results are averaged over 24 different seeds.

The reward function used is mostly the same as in the Gymnasium version, except instead of terminating on game over or the lander not being awake (i.e. landed), a -1 or +1 reward is issued each timestep respectively. Note that as seen in figs. 2 to 4, this can lead to very large negative rewards.

### B.4  ANT

V4 version with `terminate_when_unhealthy=False` so that there are more maximum length trajectories. Trajectories are 1k timesteps and preference fragments are 50 timesteps. Results are averaged over 12 different seeds.

## C  SUPPLEMENTARY RESULTS

| Method | RM epochs per iter | Final Ground Truth Reward ± std error | | | |
|---|---|---|---|---|---|
| | | Half Cheetah | Cliff Walking | Lunar Lander | Ant |
| LEOPARD (ours) | - | **1460** ± 228 | **763** ± 118 | **-231** ± 138 | -382 ± 303 |
| AILP | - | -91 ± 20 | 678 ± 167 | -2271 ± 421 | **220** ± 151 |
| DeepIRL then RLHF | 1 | 511 ± 118 | 113 ± 184 | -1565 ± 212 | -1733 ± 159 |
| DeepIRL then RLHF | 2 | 492 ± 159 | 79 ± 188 | -1652 ± 236 | -1539 ± 213 |
| DeepIRL then RLHF | 4 | 1269 ± 208 | 33 ± 144 | -748 ± 149 | -1522 ± 192 |
| DeepIRL then RLHF | 8 | 718 ± 176 | 98 ± 168 | -1292 ± 282 | -1337 ± 216 |
| RLHF then DeepIRL | 1 | 156 ± 138 | - | - | - |
| RLHF then DeepIRL | 2 | 229 ± 228 | - | - | - |
| RLHF then DeepIRL | 4 | 769 ± 242 | - | - | - |
| RLHF then DeepIRL | 8 | 599 ± 196 | - | - | - |
| LEOPARD (ours) | - | **797** ± 242 | 667 ± 120 | **-201** ± 147 | -439 ± 157 |
| AILP | - | -102 ± 23 | 536 ± 141 | -376 ± 125 | -396 ± 139 |
| DeepIRL | 1 | 31 ± 170 | 472 ± 161 | -1154 ± 221 | -698 ± 299 |
| DeepIRL | 2 | 205 ± 146 | 810 ± 162 | -664 ± 172 | -1303 ± 229 |
| DeepIRL | 4 | 661 ± 180 | 737 ± 107 | -1140 ± 230 | **-271** ± 476 |
| DeepIRL | 8 | 385 ± 159 | **977** ± 78 | -720 ± 229 | -827 ± 213 |

Table 2: Final ground truth reward with standard error for LEOPARD against a variety of baselines. (Top) 50/50 mix of preferences and positive demonstrations with baselines of AILP, performing DeepIRL followed by RLHF, and performing RLHF followed by DeepIRL (Half Cheetah only). See Figure 2 for reward vs algorithm iteration. (Bottom) Only positive demonstrations with baselines of AILP and DeepIRL. See Figure 3 for reward vs algorithm iteration. 'RM epochs per iter' is the number of training epochs for the reward model on each iteration of the algorithm, required to be fixed for DeepIRL. **Best** in column for section.

| Feedback types | Final Ground Truth Reward ± std error | | | |
| --- | --- | --- | --- | --- |
| | Half Cheetah | Cliff Walking | Lunar Lander | Ant |
| Preferences | $1225 \pm 219$ | $289 \pm 147$ | $-213 \pm 110$ | $-980 \pm 242$ |
| Positive demonstrations | $797 \pm 242$ | $667 \pm 120$ | $-201 \pm 147$ | $-439 \pm 157$ |
| Preferences and positive demos | $\mathbf{1460} \pm 228$ | $763 \pm 118$ | $-232 \pm 138$ | $\mathbf{-383} \pm 303$ |
| Positive and negative demos | $1072 \pm 206$ | $792 \pm 104$ | $\mathbf{-67} \pm 81$ | $-2598 \pm 44$ |
| Prefs, pos and neg demos | $1097 \pm 183$ | $\mathbf{1015} \pm 30$ | $-182 \pm 110$ | $-2463 \pm 69$ |

Table 3: Final ground truth reward with standard error for LEOPARD across a variety of mixture of types of feedback. For details on feedback amounts per environment and the reward vs algorithm iteration see Figure 4. **Best** in column.

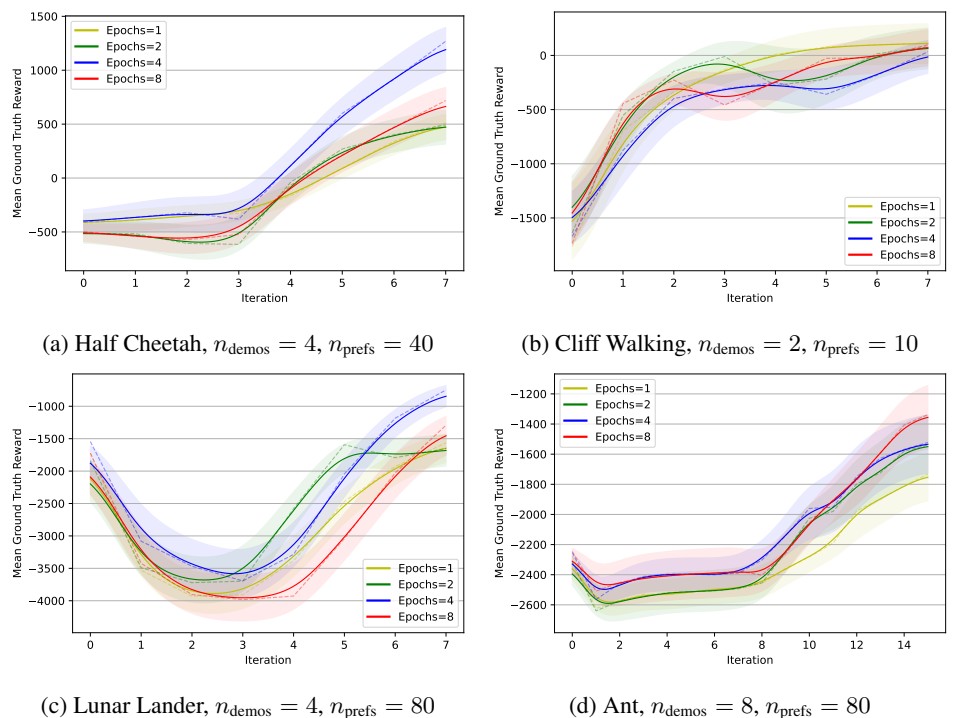

(a) Half Cheetah, $n_{\text{demos}} = 4$, $n_{\text{prefs}} = 40$

(b) Cliff Walking, $n_{\text{demos}} = 2$, $n_{\text{prefs}} = 10$

(c) Lunar Lander, $n_{\text{demos}} = 4$, $n_{\text{prefs}} = 80$

(d) Ant, $n_{\text{demos}} = 8$, $n_{\text{prefs}} = 80$

Figure 7: Breakdown of the DeepIRL followed by RLHF baseline, for different numbers of epochs that the reward model was trained for per algorithm iteration. The lines denote the mean of the ground truth reward function, with shaded standard errors, against algorithm iterations. Solid lines are smoothed means for clarity, dashed lines give raw values.

| Method | RM epochs per iter | Cliff Walking Outliers |
| --- | --- | --- |
| LEOPARD (ours) | - | 0 |
| AILP (demonstrations and preferences) | - | 4 |
| AILP (demonstrations only) | - | 0 |
| DeepIRL only | 1, 2, 4, 8 | 0 |
| DeepIRL then RLHF | 1 | 5 |
| DeepIRL then RLHF | 2 | 7 |
| DeepIRL then RLHF | 4 | 2 |
| DeepIRL then RLHF | 8 | 4 |

Table 4: Outliers for Cliff Walking that were removed from the main analysis. This is defined as having less than -3000 reward on any iteration from the second onwards. Note there were 16 random seeds in total. Values for LEOPARD and DeepIRL only given as a total across all relevant experiments.

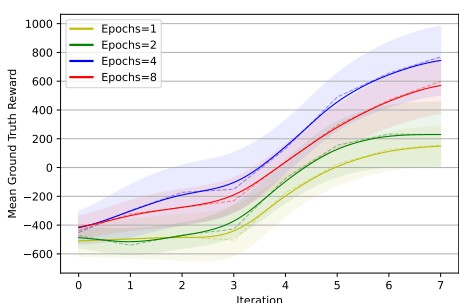

Figure 8: Breakdown of the RLHF followed by DeepIRL baseline for Half Cheetah ($n_{\text{demos}} = 4$, $n_{\text{prefs}} = 40$), for different numbers of epochs that the reward model was trained for per algorithm iteration. The lines denote the mean of the ground truth reward function, with shaded standard errors, against algorithm iterations. Solid lines are smoothed means for clarity, dashed lines give raw values.

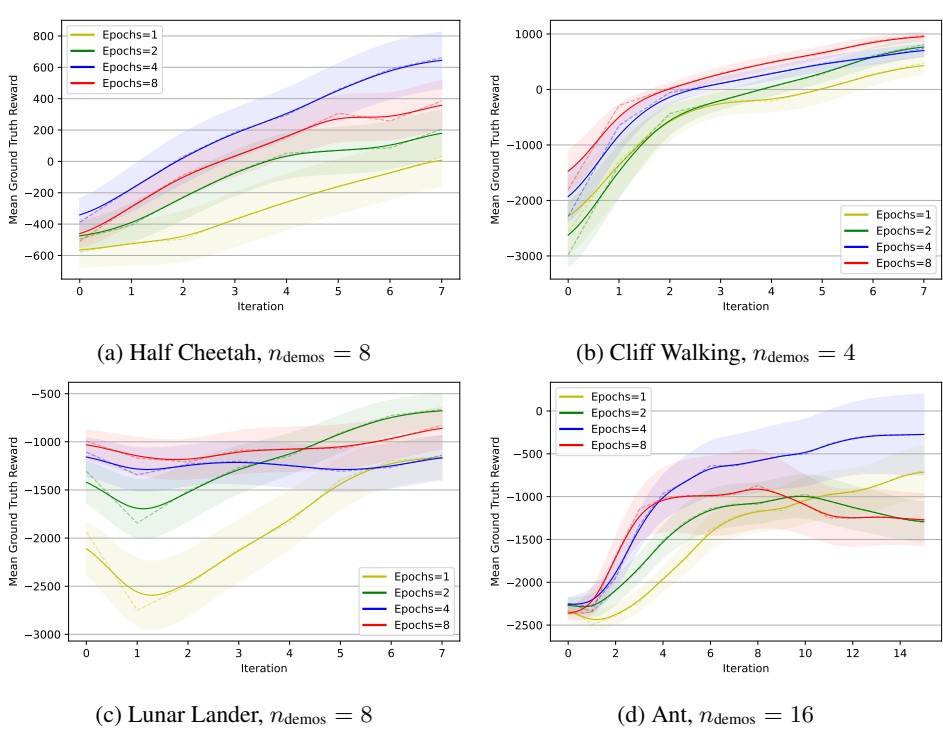

Figure 9: Breakdown of the DeepIRL baseline, for different numbers of epochs that the reward model was trained for per algorithm iteration. The lines denote the mean of the ground truth reward function, with shaded standard errors, against algorithm iterations. Solid lines are smoothed means for clarity, dashed lines give raw values.

# D    MAIN PROOFS

Here we more stringently define and prove the theoretical result from the end of section 3.1, and then prove the models considered in Appendix E do not satisfy it.

**Theorem 1.** *Upper bounds on RRPO loss give lower bounds on reward difference of related fragments. For all $\epsilon > 0$, if $\mathcal{L}_{RRPO} \leq \epsilon$, then for all $\tau_a, \tau_b \in \mathcal{D}^2$ where there exists a $<_x \in \mathcal{C}$ such that $\tau_a <_x \tau_b$, we have the following:*

$$R_\theta(\tau_b) - R_\theta(\tau_a) > -\frac{1}{\beta_x} \log(e^\epsilon - 1), \tag{9}$$

*where $\beta_x$ is the rationality coefficient of $<_x$.*

*Proof.* We will prove this by contrapositive, that is if:

$$R_\theta(\tau_b) - R_\theta(\tau_a) \leq -\frac{1}{\beta_x} \log(e^\epsilon - 1), \tag{10}$$

for some $\epsilon > 0$, and there exists a $<_x$ such that $\tau_a <_x \tau_b$, then $\mathcal{L}_{\text{RRPO}} > \epsilon$.

Assume eq. (10) and that the relevant $<_x$ exists. Consider eq. (6):

$$\mathcal{L}_{\text{RRPO}}(\theta) = -\log P_{\text{RRPO}}(\mathcal{C}|\mathcal{D}, \theta)$$

$$= -\sum_{(\tau_i, <_j) \in \mathcal{D} \times \mathcal{C}} \log \frac{\exp(\beta_j R_\theta(\tau_i))}{\exp(\beta_j R_\theta(\tau_i)) + \sum_{\tau_k \in \mathcal{D}} \mathbf{1}_{\tau_k <_j \tau_i} \exp(\beta_j R_\theta(\tau_k))}$$

$$= \sum_{(\tau_i, <_j) \in \mathcal{D} \times \mathcal{C}} \log \frac{\exp(\beta_j R_\theta(\tau_i)) + \sum_{\tau_k \in \mathcal{D}} \mathbf{1}_{\tau_k <_j \tau_i} \exp(\beta_j R_\theta(\tau_k))}{\exp(\beta_j R_\theta(\tau_i))}$$

$$= \sum_{(\tau_i, <_j) \in \mathcal{D} \times \mathcal{C}} \log \left( 1 + \frac{\sum_{\tau_k \in \mathcal{D}} \mathbf{1}_{\tau_k <_j \tau_i} \exp(\beta_j R_\theta(\tau_k))}{\exp(\beta_j R_\theta(\tau_i))} \right).$$

Consider the term $(\tau_b, <_x)$, and bring it outside the summation.

$$\mathcal{L}_{\text{RRPO}}(\theta) = \log \left( 1 + \frac{\sum_{\tau_k \in \mathcal{D}} \mathbf{1}_{\tau_k <_x \tau_b} \exp(\beta_x R_\theta(\tau_k))}{\exp(\beta_x R_\theta(\tau_b))} \right) + \sum_{\substack{(\tau_i, <_j) \in \mathcal{D} \times \mathcal{C} \\ (\tau_i, <_j) \neq (\tau_b, <_x)}} \log \left( 1 + ... \right).$$

The remaining terms are strictly positive, and $\mathbf{1}_{\tau_a <_x \tau_b} = 1$.

$$\mathcal{L}_{\text{RRPO}}(\theta) > \log \left( 1 + \frac{\exp(\beta_x R_\theta(\tau_a)) + ...}{\exp(\beta_x R_\theta(\tau_b))} \right)$$

$$= \log \left( 1 + \exp(\beta_x R_\theta(\tau_a) - \beta_x R_\theta(\tau_b)) + \frac{...}{\exp(\beta_x R_\theta(\tau_b))} \right)$$

$$> \log \left( 1 + \exp(\beta_x (R_\theta(\tau_a) - R_\theta(\tau_b))) \right),$$

by ignoring terms that are strictly positive. Sub in eq. (10).

$$\mathcal{L}_{\text{RRPO}}(\theta) > \log \left( 1 + \exp \left( \beta_x \left( \frac{1}{\beta_x} \log(e^\epsilon - 1) \right) \right) \right)$$

$$= \log \left( 1 + e^\epsilon - 1 \right)$$

$$= \epsilon,$$

as required.  □

Consider a special case where $\epsilon = \log 2$, eq. (9) becomes:

$$R_\theta(\tau_b) - R_\theta(\tau_a) > -\frac{1}{\beta_x} \log \left( e^{\log 2} - 1 \right)$$

$$= 0,$$

$$\therefore R_\theta(\tau_b) > R_\theta(\tau_a).$$

# E   ALTERNATIVE RRC-DERIVED APPROACHES

RRPO and LEOPARD are very simple and natural extensions of existing work, however, they are not trivially so. Building off RRC, there are several approaches to preference and demonstration learning that appear natural and are simple, and yet are deficient. Here we explore two of them in the preference and ranked positive demonstrations only setting.

Let the notation be as defined in section 3.2. We will assume that preferences, positive demonstration selection, and the rankings over the positive demonstrations are all independent. Our overall likelihood function shall be:

$$
\begin{aligned}
P_{\text{Feedback}}(\mathcal{C}|\mathcal{D},\theta) = {} & P_{\text{Pos-Demo}}(\mathcal{D}_{\text{pos}} \succ \mathcal{D}_{\text{agent}}|\mathcal{D}_{\text{pos}},\mathcal{D}_{\text{agent}},\theta) \\
& \cdot P_{\text{Rank}}(<_{\text{pos}}|\mathcal{D}_{\text{pos}},\theta) \\
& \cdot \prod_{(\tau_a,\tau_b)\in\mathcal{P}} P_{\text{RLHF}}(\tau_a \succ \tau_b|\theta),
\end{aligned}
\tag{11}
$$

where $P_{\text{Rank}}$ is something sensible.

We consider two potential candidates for $P_{\text{Pos-Demo}}$ derived via RRC in a simple manner:

$$
P_{\text{Sum-of-Choices}}(...) = \sum_{\tau\in\mathcal{D}_{\text{pos}}} P_{\text{RRC}}(C_\tau|\mathcal{D}_{\text{pos}}\cup\mathcal{D}_{\text{agent}},\theta),
\tag{12}
$$

$$
P_{\text{Choose-Best-Average}}(...) = P_{\text{RRC}}(C_{\text{Avg}(\mathcal{D}_{\text{pos}})}|\{\text{Avg}(\mathcal{D}_{\text{pos}}),\text{Avg}(\mathcal{D}_{\text{agent}})\},\theta).
\tag{13}
$$

Thus:

$$
P_{\text{Sum-of-Choices}}(...) = \frac{\sum_{\tau\in\mathcal{D}_{\text{pos}}}\exp(R_\theta(\tau))}{\sum_{\tau\in\mathcal{D}_{\text{pos}}}\exp(R_\theta(\tau)) + \sum_{\tau\in\mathcal{D}_{\text{agent}}}\exp(R_\theta(\tau))},
\tag{14}
$$

$$
P_{\text{Choose-Best-Average}}(...) = \frac{\exp\left(\frac{1}{|\mathcal{D}_{\text{pos}}|}\sum_{\tau\in\mathcal{D}_{\text{pos}}}R_\theta(\tau)\right)}{\exp\left(\frac{1}{|\mathcal{D}_{\text{pos}}|}\sum_{\tau\in\mathcal{D}_{\text{pos}}}R_\theta(\tau)\right) + \exp\left(\frac{1}{|\mathcal{D}_{\text{agent}}|}\sum_{\tau\in\mathcal{D}_{\text{agent}}}R_\theta(\tau)\right)},
\tag{15}
$$

with

$$
\mathcal{L}_{\text{SoC}} = -\log P_{\text{Sum-of-Choices}},
\tag{16}
$$

$$
\mathcal{L}_{\text{CBA}} = -\log P_{\text{Choose-Best-Average}}.
\tag{17}
$$

Rationality coefficients are omitted since they are not critical to this analysis. We shall show that these models have undesirable theoretical properties, and poorer empirical performance compared to LEOPARD.

## E.1   THEORETICAL PROPERTIES

Neither $P_{\text{Sum-of-Choices}}$ nor $P_{\text{Choose-Best-Average}}$ have the property that upper bounds on their negative-log-likelihood give rise to lower bounds on reward differences between demonstrated trajectories and ones sampled from the agent, unlike $P_{\text{RRPO}}$. We prove this in theorems 2 and 3 in Appendix E.2.1. Whilst this may not seem too critical, its combination with the potential effects of $P_{\text{Rank}}$, and its interaction with exploration in RL, can cause a very undesirable failure mode.

Imagine an environment where three distinct behaviours are possible, A, B, and C. We prefer C to B, and B to A, so we provide a demonstration of B and C each, $\tau_b, \tau_c$, and express via the ranking model that $\tau_c \succ \tau_b$. This ranking is fitted by assigning high reward to C, and low to B. Our agent is initialised generating from A. Our demonstration model, seeing $\tau_c$ have high reward, does not lower the reward of A that much, and does not mind that $\tau_b$ has low reward. We're left with low loss and yet a reward model that could prefer A to B.

Now consider that our environment has some unfavourable dynamics. Policies that generate A, are quite different from those that generate C, with B being somewhere between the two. Thus, to eventually generate C, our policy will first need to explore B. However, our reward model gives it

lower reward when it tries this, and so the agent sticks to what it thinks is best, behaviour A, much to our disappointment.

Whilst a little contrived, the above story highlights a certain failure mode that could occur if one combined demonstration rankings with a demonstration model that does not satisfy theorem 1. If it did satisfy it, such as for RRPO and LEOPARD, then low loss cannot be achieved unless the reward model prefers B to A, preventing the issue.

Alleviating this problem by omitting the rankings is suboptimal, as we lose information. However, $P_{\text{Sum-of-Choices}}$ suffers further. It is shown in Appendix E.2.2 that the gradient of $\mathcal{L}_{\text{SoC}}$ with respect to $\theta$ can be expressed in the following form.

$$-\frac{\partial}{\partial \theta}\mathcal{L}_{\text{SoC}} = \sum_{\tau_a \in \mathcal{D}_{\text{agent}}} P_{\text{RRC}}(C_a|\mathcal{T},\theta) \left( \sum_{\tau_p \in \mathcal{D}_{\text{pos}}} P_{\text{RRC}}(C_p|\mathcal{D}_{\text{pos}},\theta)\frac{\partial}{\partial \theta}R_\theta(\tau_p) - \frac{\partial}{\partial \theta}R_\theta(\tau_a) \right), \quad (18)$$

where $C_i$ is the human choice for $\tau_i$, and $\mathcal{T} = \mathcal{D}_{\text{pos}} \cup \mathcal{D}_{\text{agent}}$. We see that the reward of agent trajectories are pushed down proportional to the probability that they would be chosen out of the combined set of trajectories. This makes sense—if our reward model thinks highly of specific agent trajectories, it ought to adjust its beliefs so that it no longer favours them.

However, the demonstration trajectories are also pushed up in reward proportional to the probability that they would be chosen. That is to say, the better the reward model thinks the demonstrated trajectory is, the more it thinks it should increase its reward, a positive feedback loop! In practice, the reward model is going to have some initial preferences over the demonstrated trajectories due to its initialisation. Since this will be random, it will most likely be incorrect. It will then proceed to reinforce its own incorrect beliefs and lock-in its own ranking of the demonstrations. This means our reward model will not provide correct rewards to guide the agent towards better behaviour in the trajectory space around the demonstrations. Furthermore, if it generalises from these incorrect beliefs, it could also become wrong about other parts of trajectory space, further reducing the quality of the reward signal for the agent.

## E.2 CHAPTER PROOFS AND DERIVATIONS

### E.2.1 REWARD BOUNDS

**Theorem 2.** *Upper bounds on Sum-of-Choices loss do not give lower bounds on reward difference between demonstrations and agent trajectories. For all $\epsilon > 0$, if $\mathcal{L}_{SoC} \leq \epsilon$, we cannot guarantee that*

$$R_\theta(\tau_p) - R_\theta(\tau_a) > f(\epsilon) \qquad (19)$$

*for all $\tau_p, \tau_a \in \mathcal{D}_{pos} \times \mathcal{D}_{agent}$, where $f$ is a function of type $\mathbb{R}^+ \to \mathbb{R}$.*

*Proof.* We will prove this by example.

Consider

$$\mathcal{D}_{\text{pos}} = \{\tau_1, \tau_2\},$$
$$\mathcal{D}_{\text{agent}} = \{\tau_a\},$$
$$R_\theta(\tau_1) = r_1,$$
$$R_\theta(\tau_2) = r_2,$$
$$R_\theta(\tau_a) = r_a.$$

We now expand eq. (16) with eq. (14) and the above.

$$\mathcal{L}_{\text{SoC}}(\theta) = -\log\left(\frac{e^{r_1} + e^{r_2}}{e^{r_1} + e^{r_2} + e^{r_a}}\right)$$

$$= \log\left(1 + \frac{e^{r_a}}{e^{r_1} + e^{r_2}}\right).$$

Assume $\mathcal{L}_{\text{SoC}} \leq \epsilon$, therefore

$$\log\left(1 + \frac{e^{r_a}}{e^{r_1} + e^{r_2}}\right) \leq \epsilon,$$

$$r_a \leq \log\left((e^\epsilon - 1)(e^{r_1} + e^{r_2})\right).$$

Let

$$r_a = \log\left((e^\epsilon - 1)(e^{r_1} + e^{r_2})\right).$$

Consider $r_1 - r_a$, substituting in the above expression:

$$\begin{aligned}
r_1 - r_a &= r_1 - \log((e^\epsilon - 1)(e^{r_1} + e^{r_2})) \\
&= r_1 - \log(e^\epsilon - 1) - \log(e^{r_1} + e^{r_2}) \\
&\leq r_1 - \log(e^\epsilon - 1) - r_2,
\end{aligned}$$

as $\log(x + y) \geq \log(y)$ for positive $x$ and $y$. Thus, we see that for a fixed $r_1$ and $\epsilon$, we can choose $r_2$ and $r_a$ such that $\mathcal{L}_{\text{SoC}} \leq \epsilon$, but $r_1 - r_a$ can be arbitrarily negative. □

**Theorem 3.** *Upper bounds on Choose-Best-Average loss do not give lower bounds on reward difference between demonstrations and agent trajectories. For all $\epsilon > 0$, if $\mathcal{L}_{CBA} \leq \epsilon$, we cannot guarantee that*

$$R_\theta(\tau_p) - R_\theta(\tau_a) > f(\epsilon) \tag{20}$$

*for all $\tau_p, \tau_a \in \mathcal{D}_{pos} \times \mathcal{D}_{agent}$, where $f$ is a function of type $\mathbb{R}^+ \to \mathbb{R}$.*

*Proof.* We will proceed similarly to the above, assuming the same notation.

Expanding eq. (17) with eq. (15).

$$\begin{aligned}
\mathcal{L}_{\text{CBA}}(\theta) &= -\log\left(\frac{\exp\left(\frac{1}{2}(r_1 + r_2)\right)}{\exp\left(\frac{1}{2}(r_1 + r_2)\right) + \exp(r_a)}\right) \\
&= \log\left(1 + \frac{\exp(r_a)}{\exp\left(\frac{1}{2}(r_1 + r_2)\right)}\right) \\
&= \log\left(1 + \exp\left(r_a - \frac{1}{2}(r_1 + r_2)\right)\right).
\end{aligned}$$

Assume $\mathcal{L}_{\text{CBA}} \leq \epsilon$, therefore

$$\log\left(1 + \exp\left(r_a - \frac{1}{2}(r_1 + r_2)\right)\right) \leq \epsilon,$$

$$r_a \leq \log(e^\epsilon - 1) + \frac{1}{2}(r_1 + r_2).$$

Let

$$r_a = \log(e^\epsilon - 1) + \frac{1}{2}(r_1 + r_2).$$

Consider $r_1 - r_a$, substituting in the above expression:

$$r_1 - r_a = r_1 - \log(e^\epsilon - 1) - \frac{1}{2}(r_1 + r_2).$$

Again, we see that for a fixed $r_1$ and $\epsilon$, we can choose $r_2$ and $r_a$ such that $\mathcal{L}_{\text{SoC}} \leq \epsilon$, but $r_1 - r_a$ can be arbitrarily negative. □

### E.2.2 LOSS GRADIENTS

Here we will show that the gradient with respect to $\theta$ of $\mathcal{L}_{\text{SoC}}$ can be expressed in the form given in eq. (18) of appendix E.1.

First we give a simplification of deterministic RRC with $\beta = 1$ and $\psi(x) = x$ for all $x$, and some additional notation:

$$C : () \rightarrow \mathcal{D},$$

$$P_{\text{RRC}}(C_i|\mathcal{D}, \theta) = \frac{e^{R_\theta(\tau_i)}}{\sum_{\tau_j \in \mathcal{D}} e^{R_\theta(\tau_j)}},$$

$$\mathcal{T} = \mathcal{D}_{\text{pos}} \cup \mathcal{D}_{\text{agent}}.$$

Now we derive some useful identities.

$$\frac{\partial}{\partial \theta} \log \sum_{\tau \in \mathcal{D}} e^{R_\theta(\tau)} = \frac{\frac{\partial}{\partial \theta} \sum_{\tau_i \in \mathcal{D}} e^{R_\theta(\tau_i)}}{\sum_{\tau_j \in \mathcal{D}} e^{R_\theta(\tau_j)}}$$

$$= \sum_{\tau_i \in \mathcal{D}} \frac{\frac{\partial}{\partial \theta} e^{R_\theta(\tau_i)}}{\sum_{\tau_j \in \mathcal{D}} e^{R_\theta(\tau_j)}}$$

$$= \sum_{\tau_i \in \mathcal{D}} \frac{e^{R_\theta(\tau_i)}}{\sum_{\tau_j \in \mathcal{D}} e^{R_\theta(\tau_j)}} \frac{\partial}{\partial \theta} R_\theta(\tau_i)$$

$$= \sum_{\tau_i \in \mathcal{D}} P_{\text{RRC}}(C_i|\mathcal{D}, \theta) \frac{\partial}{\partial \theta} R_\theta(\tau_i), \tag{21}$$

$$P_{\text{RRC}}(C_i|\mathcal{A}, \theta) = \frac{e^{R_\theta(\tau_i)}}{\sum_{\tau_j \in \mathcal{A}} e^{R_\theta(\tau_j)}}$$

$$= \frac{e^{R_\theta(\tau_i)}}{\sum_{\tau_j \in \mathcal{A}} e^{R_\theta(\tau_j)}} \frac{\sum_{\tau_k \in \mathcal{A} \cup \mathcal{B}} e^{R_\theta(\tau_k)}}{\sum_{\tau_k \in \mathcal{A} \cup \mathcal{B}} e^{R_\theta(\tau_k)}}$$

$$= \frac{P_{\text{RRC}}(C_i|\mathcal{A} \cup \mathcal{B}, \theta)}{\sum_{\tau_j \in \mathcal{A}} P_{\text{RRC}}(C_j|\mathcal{A} \cup \mathcal{B}, \theta)}, \tag{22}$$

$$P_{\text{RRC}}(C_i|\mathcal{A}, \theta) - P_{\text{RRC}}(C_i|\mathcal{A} \cup \mathcal{B}, \theta) = \frac{P_{\text{RRC}}(C_i|\mathcal{A} \cup \mathcal{B}, \theta)}{\sum_{\tau_j \in \mathcal{A}} P_{\text{RRC}}(C_j|\mathcal{A} \cup \mathcal{B}, \theta)} - P_{\text{RRC}}(C_i|\mathcal{A} \cup \mathcal{B}, \theta)$$

$$= \frac{P_{\text{RRC}}(C_i|\mathcal{A} \cup \mathcal{B}, \theta) \left(1 - \sum_{\tau_j \in \mathcal{A}} P_{\text{RRC}}(C_i|\mathcal{A} \cup \mathcal{B}, \theta)\right)}{\sum_{\tau_j \in \mathcal{A}} P_{\text{RRC}}(C_j|\mathcal{A} \cup \mathcal{B}, \theta)}$$

$$= \frac{P_{\text{RRC}}(C_i|\mathcal{A} \cup \mathcal{B}, \theta) \sum_{\tau_k \in \mathcal{B}} P_{\text{RRC}}(C_k|\mathcal{A} \cup \mathcal{B}, \theta)}{\sum_{\tau_j \in \mathcal{A}} P_{\text{RRC}}(C_j|\mathcal{A} \cup \mathcal{B}, \theta)}$$

$$= \sum_{\tau_k \in \mathcal{B}} P_{\text{RRC}}(C_k|\mathcal{A} \cup \mathcal{B}, \theta) \frac{P_{\text{RRC}}(C_i|\mathcal{A} \cup \mathcal{B}, \theta)}{\sum_{\tau_j \in \mathcal{A}} P_{\text{RRC}}(C_j|\mathcal{A} \cup \mathcal{B}, \theta)}$$

$$= \sum_{\tau_k \in \mathcal{B}} P_{\text{RRC}}(C_k|\mathcal{A} \cup \mathcal{B}, \theta) P_{\text{RRC}}(C_i|\mathcal{A}, \theta) \tag{23}$$

Now we use these identities to derive the special form of the gradient of $\mathcal{L}_{\text{SoC}}$.

$$
\begin{aligned}
-\frac{\partial}{\partial\theta}\mathcal{L}_{\text{SoC}} =& \frac{\partial}{\partial\theta}\log\frac{\sum_{\tau\in\mathcal{D}_{\text{pos}}}e^{R_\theta(\tau)}}{\sum_{\tau\in\mathcal{D}_{\text{pos}}}e^{R_\theta(\tau)}+\sum_{\tau\in\mathcal{D}_{\text{agent}}}e^{R_\theta(\tau)}} \\
=& \frac{\partial}{\partial\theta}\log\sum_{\tau\in\mathcal{D}_{\text{pos}}}e^{R_\theta(\tau)}-\frac{\partial}{\partial\theta}\log\sum_{\tau\in\mathcal{T}}e^{R_\theta(\tau)} \\
=& \sum_{\tau_p\in\mathcal{D}_{\text{pos}}}P_{\text{RRC}}(C_p|\mathcal{D}_{\text{pos}},\theta)\frac{\partial}{\partial\theta}R_\theta(\tau_p)-\sum_{\tau_i\in\mathcal{T}}P_{\text{RRC}}(C_i|\mathcal{T},\theta)\frac{\partial}{\partial\theta}R_\theta(\tau_i) \\
=& \sum_{\tau_p\in\mathcal{D}_{\text{pos}}}P_{\text{RRC}}(C_p|\mathcal{D}_{\text{pos}},\theta)\frac{\partial}{\partial\theta}R_\theta(\tau_p)-\sum_{\tau_p\in\mathcal{D}_{\text{pos}}}P_{\text{RRC}}(C_p|\mathcal{T},\theta)\frac{\partial}{\partial\theta}R_\theta(\tau_p) \\
& -\sum_{\tau_a\in\mathcal{D}_{\text{agent}}}P_{\text{RRC}}(C_a|\mathcal{T},\theta)\frac{\partial}{\partial\theta}R_\theta(\tau_a) \\
=& \sum_{\tau_p\in\mathcal{D}_{\text{pos}}}\left(P_{\text{RRC}}(C_p|\mathcal{D}_{\text{pos}},\theta)-P_{\text{RRC}}(C_p|\mathcal{T},\theta)\right)\frac{\partial}{\partial\theta}R_\theta(\tau_p) \\
& -\sum_{\tau_a\in\mathcal{D}_{\text{agent}}}P_{\text{RRC}}(C_a|\mathcal{T},\theta)\frac{\partial}{\partial\theta}R_\theta(\tau_a) \\
=& \sum_{\tau_p\in\mathcal{D}_{\text{pos}}}\sum_{\tau_a\in\mathcal{D}_{\text{agent}}}P_{\text{RRC}}(C_a|\mathcal{T},\theta)P_{\text{RRC}}(C_p|\mathcal{D}_{\text{pos}},\theta)\frac{\partial}{\partial\theta}R_\theta(\tau_p) \\
& -\sum_{\tau_a\in\mathcal{D}_{\text{agent}}}P_{\text{RRC}}(C_a|\mathcal{T},\theta)\frac{\partial}{\partial\theta}R_\theta(\tau_a) \\
=& \sum_{\tau_a\in\mathcal{D}_{\text{agent}}}P_{\text{RRC}}(C_a|\mathcal{T},\theta)\left(\sum_{\tau_p\in\mathcal{D}_{\text{pos}}}P_{\text{RRC}}(C_p|\mathcal{D}_{\text{pos}},\theta)\frac{\partial}{\partial\theta}R_\theta(\tau_p)-\frac{\partial}{\partial\theta}R_\theta(\tau_a)\right). \quad (24)
\end{aligned}
$$

