# OpenReview forum: "Learning from Preferences and Mixed Demonstrations in General Settings"
_ICLR.cc/2025/Conference — Submitted to ICLR 2025_

### Official Review · Reviewer_uaHY · 2024-11-01

**Soundness:** 2
**Presentation:** 2
**Contribution:** 2
**Rating:** 5
**Confidence:** 3

**Summary:**

In this paper, the authors present LEOPARD, a novel method for learning from preferences and mixed demonstrations. LEOPARD is built upon the proposed framework that models human feedback as reward-rational partial orderings over trajectories. The method's effectiveness is evaluated across three diverse tasks, encompassing both discrete and continuous observation and action spaces.

**Strengths:**

Learning from preferences and mixed demonstrations addresses a important and highly relevant problem for the community.

**Weaknesses:**

- **Lack of Baseline Comparisons**: The paper does not provide comparisons to existing preference learning methods. Incorporating baselines from recent work, such as [1][2][3], would strengthen the empirical evaluation.
- **Simplistic Environments**: The evaluation is conducted on relatively simple environments. In contrast, other preference learning studies have utilized a broader range of tasks, such as Meta-World[4] and [5], which could better demonstrate the robustness of the method.
- **Unclear Novelty in Relation to RRL[6]**: The authors claim that their Reward-Rational Partial Orderings framework is a novel advancement compared to RRP. However, the novelty is not convincingly articulated, and the distinction remains unclear when directly compared to the RRP paper.
- **Writing and Clarity**: The paper’s writing can be improved in several areas:
    - The abstract does not provide a clear explanation of how LEOPARD works. Mentioning the concept of reward-rational partial orderings upfront would give readers a better understanding.
    - Equations, such as Eq. 1 and 2, are not well integrated into the text, making it difficult to follow the technical details.
    - Including a comprehensive illustrative figure to outline the entire algorithm would enhance clarity and reader comprehension.
    - The description of algorithm iterations in Figures 1-3 is ambiguous, and it is unclear what exactly they represent. Providing more detail or clearer labels would be helpful.

[1Hejna, J., Rafailov, R., Sikchi, H., Finn, C., Niekum, S., Knox, W. B., & Sadigh, D. (2024). *Contrastive Preference Learning: Learning From Human Feedback without RL*. In *Proceedings of the International Conference on Learning Representations (ICLR) 2024*

[2]Kim, C., Park, J., Shin, J., Lee, H., Abbeel, P., & Lee, K. (2023). *Preference Transformer: Modeling Human Preferences using Transformers for RL*. In *Proceedings of the International Conference on Learning Representations (ICLR) 2023*

[3]Taranovic, A., Kupcsik, A. G., Freymuth, N., & Neumann, G. (2023). *Adversarial Imitation Learning with Preferences*. In *Proceedings of the Eleventh International Conference on Learning Representations (ICLR) 2023*

[4]https://github.com/Farama-Foundation/Metaworld

[5]Knox, W. B., Hatgis-Kessell, S., Booth, S., Niekum, S., Stone, P., & Allievi, A. (2023). Models of Human Preference for Learning Reward Functions. ArXiv preprint

[6] Jeon, H. J., Milli, S., & Dragan, A. (2024). *Reward-Rational (Implicit) Choice: A Unifying Formalism for Reward Learning*. In *Advances in Neural Information Processing Systems (NeurIPS)*

**Questions:**

Could you please further clarify the proposed differences to RRL[1]?

Will authors make the code available?

[1]Jeon, H. J., Milli, S., & Dragan, A. (2024). *Reward-Rational (Implicit) Choice: A Unifying Formalism for Reward Learning*. In *Advances in Neural Information Processing Systems (NeurIPS)*

---

> ### Author Response · Authors · 2024-11-22
>
> Thank you for your review and for highlighting the importance of this area of research.
>
> We are grateful for you highlighting areas in the paper where we can improve the clarity and quality of our explanations. These suggestions will be addressed in our next draft. Additionally, we will include how LEOPARD performs against another, more modern, preference and demonstration combining algorithm, AILP[1], in both the settings of positive demonstrations, and positive demonstrations + preferences. We will also include results from the Mujoco Ant environment, which is more complex than the others we have tested on.
>
> To clarify the relationship between RRPO and reward-rational choice (RRC)[1], we believe that despite the strong resemblance between them, RRPO is a distinct mathematical and conceptual framework. RRC frames each piece of human feedback as a single choice from a set, whereas RRPO treats it as a partial ordering over this set, which is strictly more expressive. Whilst one can theoretically carefully decompose these partial orderings into sets of single choices, this misses the conceptual power and simplicity of RRPO.
>
> In our opinion, it is much more natural and clear how one could effectively encode different types of feedback under RRPO than RRC, when there’s no obvious single choice. It makes LEOPARD easy to express and understand, rather than obfuscating how it operates (as might happen if it was derived under RRC). This is especially apparent when comparing to the formulation of demonstrative feedback considered in the original RRC paper. Looking at that, it’s difficult to see how one might extend it to cover rankings and negative demonstrations. Our new draft will include an appendix chapter that details two alternative methods that one might straightforwardly derive from RRC to do positive-ranked-demonstration and preference learning, and how these are theoretically deficient.
>
> On the mathematical side, the framing of RRPO made it much easier for us to spot and prove theorem 1 of the paper, and will potentially elucidate further interesting theoretical results.
>
> We will make code available on github shortly after the rebuttal period ends.
>
> [1] Jeon, H. J., Milli, S., & Dragan, A. (2024). Reward-Rational (Implicit) Choice: A Unifying Formalism for Reward Learning. In Advances in Neural Information Processing Systems (NeurIPS)

---

> > ### Comment · Reviewer_uaHY · 2024-11-29
> >
> > Dear authors, thank you for your clarifications and the improvements of the manuscripts. The additional baselines and tasks have improved the quality of the submitted work, so I would increase the score to 5. However, additional evaluations would be required, especially since the performance LEOPARD is not convincing, and it should still be evaluated on more complicated tasks as mentioned in the original review.

---

> > > ### Author Response · Authors · 2024-12-01
> > >
> > > Thank you for your feedback.

---

### Official Review · Reviewer_NGe2 · 2024-11-05

**Soundness:** 3
**Presentation:** 3
**Contribution:** 1
**Rating:** 3
**Confidence:** 3

**Summary:**

This paper introduces LEOPARD: Learning Estimated Objectives from Preferences And Ranked Demonstrations. LEOPARD is a method from learning from diverse types of feedback including negative or failed demonstrations, rankings, and positive demonstrations. LEOPARD is based on a new framework that this paper introduces called reward-rational partial ordering (RRPO). LEOPARD is primarily compared against methods that pretrain through IRL on demonstrations and then finetune on preferences. The comparisons occur in 3 environments: HalfCheetah, Cliff Walking, and Lunar Lander.

**Strengths:**

I think the paper is clear, well-written, and to my knowledge, technically correct. The paper also is thoughtful in its experimentation, mostly recognizes the prior literature, and does not overclaim.

**Weaknesses:**

- I do think a preference-based IRL baseline rather than DeepIRL + finetuning can bolster the paper. I realize that preference-based inverse RL baselines aren't easy to compare against. However, nonetheless I think it can improve the paper.
- Unfortunately, only 3 environments are tested, and the results are not that strong. In HalfCheetah the confidence intervals overlap in Figure 1. Moreover, it seems the benefit primarily lies when both demonstrations and preferences are available, when the baseline methods do IRL then finetuning. To a certain degree, their method is built to be successful in this scenario. Of course, it is a good sanity check, but nonetheless limits the scope of the results.
- Another concern of mine is the scalability of the method given the relatively small environments. HalfCheetah, Lunar Lander, and Cliff Walking are not the most testing environments. Preference-based IRL methods are tricky and finicky to get to work, and doing so is part of the contributions of such papers. Unfortunately it does beg the question of whether it does scale, and what challenges LEOPARD may face when scaling.
- I appreciate the authors' noting the limitations of their work, but I do feel some of these limitations should be addressed by the paper itself to meet the bar of ICLR. In particular testing on a few more environments that are more challenging would bolster these results.

**Questions:**

NA

---

> ### Author Response · Authors · 2024-11-22
>
> Thank you for your review and for highlighting some of the strengths of our paper.
>
> To address some of the weaknesses you have raised, as part of our planned re-draft we will also include how it performs against another, more modern, preference and demonstration combining algorithm, AILP[1]. This will be done for both the settings of positive demonstrations, and positive demonstrations + preferences. We will also include results from the Mujoco Ant environment.
>
> For clarification, the intention of LEOPARD is to be a novel algorithm for learning from both preferences and mixed demonstrations, and thus its performance in these settings is of most interest to us. We see its ability to achieve performance comparable to, or better than, existing standard IRL algorithms when only (positive) demonstration data is available as an added bonus.
>
> [1] Taranovic, A., Kupcsik, A. G., Freymuth, N., & Neumann, G. (2023). Adversarial Imitation Learning with Preferences. In Proceedings of the Eleventh International Conference on Learning Representations (ICLR) 2023

---

> > ### Comment · Reviewer_NGe2 · 2024-11-25
> > **Reviewer Response to Authors**
> >
> > I acknowledge that I had read the author response, and I maintain my score.

---

> > > ### Author Response · Authors · 2024-12-01
> > >
> > > Dear reviewer,
> > >
> > > Please could you consider the updated draft as it addresses several of your concerns.
> > > Details on what has been changed have been summarised in a comment above.

---

### Official Review · Reviewer_sr9N · 2024-11-07

**Soundness:** 2
**Presentation:** 2
**Contribution:** 1
**Rating:** 6
**Confidence:** 3

**Summary:**

This paper introduces LEOPARD, a new algorithm for reinforcement learning from mixed types of feedback. This approach is based on the theoretical framework of reward-rational (implicit) choice (RRC). Based on that, they develop a general theoretical framework, namely reward-rational partial ordering (RRPO). Then it proposed an algorithm to construct preference rankings from positive, negative and preference ranking data and then feed it to the RRPO objective, namely the LEOPARD method. Through experiments on three environments from the Gymnasium, it demonstrates that LEOPARD outperforms traditional methods that sequentially apply IRL on demonstrations and RLHF on preferences.

**Strengths:**

1. The RRPO framework allows it to handle partial rankings across multiple types of feedback.
2. The author shows that RRPO faithfully represents the partial orderings in appendix D.

**Weaknesses:**

1. The actual algorithm is a simple combination of well-known existing approaches. Though the RRC and RRPO framework is very general. The proposed method LEOPARD only use trajectory-level pairwise preference rankings. LEOPARD is essentially just RLHF with some synthetic data augmentation when positive and negative demonstration data are available.

2. The experiment is limited to 3 Gymnasium environments. These domains are relatively low-dimensional and may not represent more complex real-world applications, such as high-dimensional robotics or language model finetuning.

**Questions:**

See weakness

---

> ### Author Response · Authors · 2024-11-22
>
> Thank you for your review and for highlighting the ability of RRPO to faithfully learn partial rankings across many types of feedback.
>
> It appears there is some confusion about the nature of LEOPARD. To clarify, it is not just a simple combination of existing approaches. Whilst it does subsume RLHF and reduce to it in the case of only preference information, when there is only demonstration data LEOPARD implements a novel inverse RL algorithm. Furthermore, it represents a novel way of unifying and combining preference and demonstration data (along with potentially many other data sources).
>
> To be more concrete, in the general case where demonstration data is available, it does not implement trajectory-level pairwise preference rankings. Instead, it implements a series of Boltzmann-Rational choices over sets containing many trajectories—note the summation in the right hand term of the denominator in equation 5. Compared to sampling pairwise preferences based on rank data, RRPO/LEOPARD can capture the full hierarchy of the human’s values over the given trajectories. The reward for each one can be increased/decreased in proportion to where it is in this hierarchy at every reward model optimisation step. For pairwise comparisons, this proportionality only arises in the limit of many optimisation steps, as when evaluating each pairwise preference the algorithm is blind to how these trajectories fit into the bigger picture.
>
> To address your concerns about our evaluation, as part of our planned re-draft we will include results from the Mujoco Ant environment.

---

> ### Comment · Reviewer_sr9N · 2024-11-22
>
> Thank you for your clarification. Two quick following-up question.
>
> Re: "Instead, it implements a series of Boltzmann-Rational choices over sets containing many trajectories—note the summation in the right hand term of the denominator in equation 5. "
>
>  - In the actual implementation of LEOPARD, do you have more than two terms in the denominator of Eq 5? How many percent of them in the whole dataset, and how significant of this part contribute to the final results?
>  - Does the D_pos or D_neg overlaps with D_pref in your implementation of LEOPARD?

---

> > ### Author Response · Authors · 2024-11-22
> >
> > Yes, in the experiments with multiple positive or negative demonstrations, there will be several terms in the (outer) summation of Eq 5 that have more than two terms in their denominator. Additionally, in the case of a single positive demonstration, the denominator for that demonstration trajectory will contain many terms for different sampled agent trajectories.
> >
> > There is no overlap between D_pos or D_neg with D_pref. D_pref solely contains preferences over the agents own behaviour. In the new draft we will have a diagram of the overall method which better illustrates this, apologies for the confusion.

---

> > > ### Comment · Reviewer_sr9N · 2024-11-25
> > >
> > > Thank you for your response. With this new explanation, I can see that my original comment "LEOPARD is essentially just RLHF with some synthetic data augmentation" and "only use trajectory-level pairwise preference rankings." are wrong. So I increased my score to 6.

---

> > > > ### Author Response · Authors · 2024-12-01
> > > >
> > > > Dear reviewer,
> > > >
> > > > Thank you for your understanding.
> > > > Additionally, please could you consider the updated draft as it addresses the other weakness you raised.
> > > > Details on what has been changed have been summarised in a comment above.

---

### Official Review · Reviewer_zpyN · 2024-11-07

**Soundness:** 2
**Presentation:** 3
**Contribution:** 2
**Rating:** 5
**Confidence:** 4

**Summary:**

This paper introduces LEOPARD, a framework capable of learning from a diverse range of data types, including preferences and ranked demonstrations. Results on the control experiment indicate that LEOPARD converges more quickly than approaches that rely solely on Inverse Reinforcement Learning (IRL) or Reinforcement Learning from Human Feedback (RLHF).

**Strengths:**

* The framework’s ability to learn from negative or failed demonstrations is intriguing.

**Weaknesses:**

* The evaluation of the proposed algorithm is limited in scope.
* The improvement achieved by LEOPARD is modest compared to the IRL method.

**Questions:**

* Could the authors provide or plot the mean ground truth reward for both the demonstration data and preference data in Figures 1 and 2? I am uncertain about the quality of the final policy, as the optimal policy for the HalfCheetah-v4 environment in Gym typically reaches a score of around 12000.

* Are there any differences between applying IRL first, followed by RLHF, and the reverse order? Could the authors also provide results comparing the policy’s quality of applying RLHF first, then IRL?

* For the stopping conditions, LEOPARD checks the value of the training loss to determine when to stop. However, if the environment changes, this threshold would also need adjustment. Are there any more effective methods to establish this threshold?

---

> ### Author Response · Authors · 2024-11-22
>
> Thank you for your review and for highlighting the flexibility of LEOPARD to learn from negative demonstrations in addition to the typical sources of feedback.
>
> To help address your concerns, we will include in the updated draft an appendix chapter with the ground truth rewards of the demonstrations. The preferences are generated over randomly selected trajectories on-policy, predominantly from the last iteration, so this can be used to infer their rough distribution of their rewards.
>
> Whilst SotA on HalfCheetah-v4 is much higher than the reward we get, we believe this is primarily because we use PPO instead of algorithms that perform highly on HalfCheetah like SAC. We did this for two reasons.
> 1) PPO converges faster (even if to a lower performance) and so it takes less computational effort to differentiate the performance of the different algorithms and data sources.
> 2) Lunar Lander and Cliff Walking (as well as LLM finetuning) have discrete action spaces and so it is typical to use PPO. By sticking with PPO we make the Half Cheetah results more comparable to these experiments and the inferences that can be made from it stronger.
>
> Testing RLHF followed by DeepIRL is an interesting idea and we will run it on Half Cheetah as an ablation. Originally we neglected it, as the other way round is seen as optimal[1], but it would be a good sanity check.
>
> For the stopping threshold, we use the same value for all environments and found that performance is not sensitive to it. Further details on this can be found in Appendix A.1.3 of the paper.
>
> [1] Erdem Bıyık, Dylan P Losey, Malayandi Palan, Nicholas C Landolfi, Gleb Shevchuk, and Dorsa Sadigh. Learning reward functions from diverse sources of human feedback: Optimally integrating demonstrations and preferences. The International Journal of Robotics Research, 41(1): 45–67, 2022.

---

> > ### Author Response · Authors · 2024-12-01
> >
> > Dear reviewer,
> >
> > Please could you consider the updated draft as it addresses some of your concerns and questions.
> > Details on what has been changed have been summarised in a comment above.

---

### Author Response · Authors · 2024-11-22

Dear Reviewers,

Thank you for taking the time to review our paper and provide us with thoughtful feedback that recognises its strengths, as well as areas we can improve.

We intend to submit an updated draft that addresses the issues you’ve highlighted. We will test LEOPARD (and baselines) on Mujoco Ant, a higher dimensional robotics task, to demonstrate its effectiveness in more complex environments. Additionally, we will include AILP[1] as an additional baseline across all environments, with both positive demonstrations, and positive demonstrations + preferences. We will also conduct an ablation study applying RLHF then DeepIRL on Half Cheetah.

Regarding the paper itself, we acknowledge that some sections may have been unclear or lacking in detail. We will update the paper to provide improved clarification on the relationship between RRPO and RRC, as well as a clearer explanation of LEOPARD, accompanied by an illustrative figure. We will also make general improvements in clarity, addressing the specific areas highlighted by your feedback.

We believe that these changes will significantly strengthen our paper and address the concerns you have raised. Thank you again for your feedback and we look forward to the opportunity to resubmit our revised paper.

[1] Taranovic, A., Kupcsik, A. G., Freymuth, N., & Neumann, G. (2023). Adversarial Imitation Learning with Preferences. In Proceedings of the Eleventh International Conference on Learning Representations (ICLR) 2023

---

> ### Author Response · Authors · 2024-11-28
> **Updated Paper**
>
> Dear Reviewers,
>
> Thank you for all your feedback. We have now uploaded the amended draft of our paper, and hope this addresses many of your remaining concerns.
>
> Major changes:
> Added results on Ant.
> Added results from AILP baseline on all environments, and RLHF then DeepIRL baseline on Half Cheetah.
> Adjusted relevant sections based on these results.
> Added illustrative diagram of overall method.
> Alternative approaches appendix chapter.
> Plotted individual settings of reward-model-training-epochs for DeepIRL-based methods in the appendix, and included only the best and average in the main figures.
>
> Minor changes:
> Included figures of demonstration trajectories in the appendix.
> Moved tables into the appendix.
> Provided additional detail to method in the abstract.
> Provided clarification on what is meant by 'algorithm iterations' in figures 1-3.
> Better integrated equations 1 and 2 into the text.
> Made some notation more consistent.

---

### Meta-Review · Area_Chair_MrxA · 2024-12-19

**Metareview:**

**Summary**: This paper introduces LEOPARD (Learning Estimated Objectives from Preferences And Ranked Demonstrations), an RL method that combines preferences and mixed demonstrations (positive and negative) to infer reward functions in general settings. The method builds on the theoretical framework of Reward-Rational Partial Orderings (RRPO), which extends Reward-Rational Choice (RRC). LEOPARD is designed to unify and optimize the use of diverse feedback types, outperforming traditional baselines like DeepIRL followed by RLHF in synthetic and continuous-control environments such as HalfCheetah, CliffWalking, and LunarLander.

**Strengths**:
- Reviewers found LEOPARD’s ability to incorporate multiple feedback types (e.g., preferences, positive/negative demonstrations, rankings) highlight practical and impactful. Reviewer `zpyN` found the flexibility to learn from negative demonstrations particularly interesting.
- Reviewers found the paper clear and well-written.

**Weaknesses**:
- Modest novelty: Reviewer `sr9N` initially expressed concerns that LEOPARD was a combination of existing techniques (which I agree with). The rebuttal clarified its unique aspects (e.g., Boltzmann-Rational choices -- although this isn't particularly novel either), but lingering doubts about its novelty remain amongst reviewers.
- Limited evaluation: All reviewers criticized the limited range of test environments. Reviewer `sr9N` and Reviewer `NGe2` pointed out that the chosen benchmarks (e.g., Gym environments) are relatively simple and may not generalize to more complex tasks, such as high-dimensional robotics or language model finetuning.
- Baseline choices: Reviewer `NGe2` questioned the absence of more modern comparison methods like preference-based IRL or Adversarial Imitation Learning with Preferences (AILP). The rebuttal partially addressed this concern by adding AILP as a baseline in the revised draft.
- Poor improvements compared to baselines: Reviewers noted that LEOPARD's performance improvements, while consistent, are modest, with overlapping confidence intervals in some cases. Reviewer `zpyN` remarked that the improvements may be specific to the feedback scenarios chosen for evaluation.

**Recommendation**: While the paper demonstrates flexibility in leveraging diverse feedback types, it falls short in experimental diversity and the strength of empirical results. Reviewers acknowledge its conceptual contribution but express concerns about its scalability and limited impact in real-world scenarios. As such, I vote to Reject this paper.

**Additional Comments On Reviewer Discussion:**

Several discussion points stuck out during the rebuttal:
- Novelty: The authors clarified LEOPARD’s differences from RLHF and its unique integration of ranked demonstrations, preferences, and Boltzmann-Rational modeling. Reviewer `sr9N` revised their opinion, increasing their score from 5 to 6. However, I still think the novelty is modest; perhaps with more impressive and extensive evaluation the paper's novelty and contributions would be more impactful.
- Evaluation scope: The authors expanded experiments to include the Mujoco Ant environment and added AILP as a baseline. These additions were appreciated but did not fully address the concerns about generalizability.
- Clarity improvements: The revised draft included clearer explanations, additional diagrams, and more rigorous algorithmic details. These changes were well-received, addressing some concerns about readability.

---

### Decision · Program_Chairs · 2025-01-22

Reject